

# High-resolution modelling of early contrail evolution from hydrogen-powered aircraft

Annemarie Lottermoser and Simon Unterstrasser

Deutsches Zentrum für Luft- und Raumfahrt, Institut für Physik der Atmosphäre, Oberpfaffenhofen, Germany

**Correspondence:** Annemarie Lottermoser (annemarie.lottermoser@dlr.de)

**Abstract.** In this study, we investigate the properties of young contrails formed behind hydrogen-powered aircraft, particularly compared to contrails from conventional kerosene combustion. High-resolution simulations of individual contrails are performed using the EULAG-LCM model, a large-eddy simulation model with fully coupled particle-based ice microphysics.

Previous studies on early contrail evolution during the vortex phase explored a range of meteorological and aircraft-related parameters but focused on contrails with ice crystal numbers and water vapor emissions typical of kerosene combustion.

This study examines the early $H_2$-contrail evolution, starting tenths of a second after exhaust emission when ice crystal formation is complete. Two key parameters are adjusted: the amount of emitted water vapor and the number of ice crystals formed during the initial stage. The emitted water vapor varies between 3.7 and 38.6 g per flight meter, depending on the fuel and aircraft type. The initial ice crystal number spans four orders of magnitude, from approximately $10^{10}$ to $10^{14}$ ice crystals per flight meter. Additionally, we extend our atmospheric scenarios to ambient temperatures up to 235 K, as $H_2$ contrails can form in warmer conditions where kerosene plumes typically cannot.

Our results show that vortex phase processes reduce the four-order magnitude difference in ice crystal number to two orders of magnitude. Moreover, relative ice crystal loss increases with increasing ambient temperatures and decreasing relative humidity levels.

Finally, we extend the parametrization of ice crystal loss from a previous study to include scenarios of contrails from hydrogen propulsion systems.

## 1 Introduction

For conventional kerosene-powered aircraft, the primary exhaust emissions consist of carbon dioxide and water vapor, with approximately 1.26 kg of water vapor emitted per kilogram of fuel burned (Bier et al., 2024). If the ambient atmosphere is sufficiently cold, water supersaturation emerges due to the mixing of the hot exhaust gases with the cold ambient air (Schumann, 1996). This happens within tenths of a second after the emission of the exhaust. Under these conditions and in the case of soot-rich combustion, water vapor condenses onto soot particles in the exhaust, forming water droplets that grow and eventually freeze into contrail ice crystals by homogeneous nucleation (Kärcher et al., 2015). In ice-supersaturated conditions, these young (i.e. several minutes old) line-shaped contrails can evolve into contrail cirrus, which may persist for several hours (Lewellen et al., 2014; Unterstrasser et al., 2017a). Contrail cirrus has been identified as a significant contributor to aviation-induced



radiative forcing. On a global scale, contrail-cirrus exert a positive radiative forcing, resulting in a net warming effect on the atmosphere (Burkhardt and Kärcher, 2011; Lee et al., 2021; Bier and Burkhardt, 2022).

Several contrail mitigation strategies are currently under debate, and their mitigation potential is estimated. Adaptations of flight routes can pursue avoiding ice-supersaturated regions, where contrail formation and persistence are likely to happen
(Gierens et al., 2020; Lee et al., 2023). Flying in formation reduces the contrail climate impact due to saturation effects (Marks et al., 2021; Unterstrasser, 2020). Moreover, modifying the fuel composition (e.g. by using sustainable aviation fuels (SAFs)) can reduce the number of emitted soot particles (Moore et al., 2017). Measurements have demonstrated that using alternative fuel blends can reduce the number of ice crystals formed as the contrail ice crystals primarily form on the emitted exhaust particles (Bräuer et al., 2021; Voigt et al., 2021). Märkl et al. (2024) find, in the specific case of the ECLIF3 experiment, a 56 %
reduction in ice particle number per mass of burnt fuel for 100 % SAF compared to a reference Jet A-1 fuel. The initial ice crystal number is the crucial quantity that controls the contrail radiative effect and life cycle for given meteorological conditions (Unterstrasser and Gierens, 2010; Burkhardt et al., 2018). A systematic reduction of the ice crystal number in contrail cirrus, e.g. triggered by using SAFs, decreases their optical depth and, consequently, their global radiative forcing (Bier et al., 2017; Burkhardt et al., 2018). However, Burkhardt et al. (2018) have found that the relationship between ice crystal number in young
contrails and the climate impact of contrail cirrus is non-linear. Consequently, recent advancements are being explored to reduce the number of emitted particles even further, such as transitioning to hydrogen-powered aircraft. As part of the ZEROe project, Airbus plans to build the first commercial hydrogen-powered aircraft by 2035 (Airbus, 2020).

Hydrogen combustion results in exhaust plumes that are significantly moister than kerosene plumes (containing approximately seven times more water vapor), and entrained atmospheric background aerosol particles are expected to serve as contrail
ice nuclei, as hydrogen exhaust plumes are void of soot particles (Kärcher et al., 2015; Bier et al., 2024). Recent research by Bier et al. (2024) indicates that hydrogen-powered aircraft could reduce the number of ice crystals formed in contrails by more than 80-90 % compared to kerosene contrails. While no soot particles are produced during hydrogen combustion, the formation of ultrafine volatile particles is possible (Ungeheuer et al., 2022), and their role in ice crystal formation is not yet understood.

Another technological option using hydrogen as an energy carrier employs a hydrogen fuel cell propulsion system (also
proposed in the ZEROe project). Hydrogen fuel cells create electrical energy that turns a propeller or fan. Fuel cells offer several advantages over conventional combustion-based technologies as they are expected to operate at a higher overall propulsion efficiency (Kazula et al., 2023). Moreover, hydrogen fuel cells eliminate emissions of carbon dioxide and nitrogen oxides. Despite ongoing development efforts, several uncertainties regarding the performance and operation of fuel cell propulsion systems remain, making this an active field of research. According to the contrail formation theory for fuel cells (Gierens,
2021), the supersaturation over water in the exhaust plume of such systems could be significantly higher than for conventional combustion systems. This leads to a much larger slope of the mixing line as described by the Schmidt-Appleman theory (Schumann, 1996), which defines the linear relationship between plume partial water vapor pressure and temperature. The plume supersaturation can attain very high values, which could trigger homogeneous droplet nucleation (HDN; Wölk and Strey, 2001) and the formation of numerous small water droplets (Jansen and Heymsfield, 2015). Given the low ambient
temperatures at typical cruise altitudes, these droplets would rapidly freeze, potentially resulting in a very high initial number



of ice crystals. In this study, we address the potential implications of this propulsion system by artificially increasing the initial number of ice crystals in our simulations. We emphasize that we do not state that fuel cell propulsion leads to a high number of contrail ice crystals. Moreover, there are design options under consideration that may avoid abundant ice crystal production. Hence, our scenarios with high ice crystal numbers should be interpreted just as one possible scenario of fuel cell propulsion.

The initial properties of contrails, such as the number of ice crystals, are crucial for evaluating the radiative properties of contrail cirrus. For a thorough assessment of contrail mitigation options that aim to reduce the number of ice crystals during the initial formation stage, it is essential to examine the implications of the formation processes and consider the subsequent evolution of the contrail during the vortex phase. The vortex phase covers the first few minutes in the contrail's lifetime. The primary process is the interaction of the contrail ice crystals with the descending counter-rotating vortex pair. This consideration

is necessary because the initial contrail ice crystal number closely matches the emitted soot particle number only under low temperature and high supersaturation conditions (Bier and Burkhardt, 2022). In contrast, significant deviations can occur under weakly supersaturated conditions and high ambient temperatures. Adiabatic heating in the descending vortex pair increases the saturation pressure, lowering the plume relative humidity. This may lead to the sublimation of the ice crystals trapped in the descending vortices (Lewellen and Lewellen, 2001; Unterstrasser and Sölch, 2010). The survival fraction of ice crystals

after the vortex phase is highly influenced by factors such as the number of nucleated ice crystals, the amount of emitted water vapor, ambient temperature, relative humidity over ice, Brunt-Väisälä frequency, and the aircraft's wingspan. Generally, the fraction of ice crystals lost during the vortex phase is larger/smaller, the more/fewer ice crystals are present in the beginning. The initial number of ice crystals is related to the aircraft's fuel flow rate and an assumed apparent ice crystal emission index $EI_{iceno}$ (Unterstrasser and Sölch, 2010; Lewellen et al., 2014). Unterstrasser (2016) states that an initial reduction of $EI_{iceno}$

from $10^{15}$ to $10^{14}\,kg^{-1}$ implies only a factor 5.3 reduction of the ice crystal number after the vortex phase. Hence, initial differences in the ice crystal number (e.g. by changes in the fuel or engine/combustor design) are reduced during the vortex phase. This means that a decreased ice crystal loss in the vortex phase partly compensates for the reduced ice nucleation (Bier and Burkhardt, 2022). The consideration of ice crystal loss processes during the vortex phase, as parametrized in Unterstrasser (2016) and implemented in larger scale models, lead to significant changes in the later contrail properties on regional (Gruber

et al., 2018) and global scale (Bier and Burkhardt, 2022).

In this study, we systematically investigate the properties of young $H_2$ contrails with a focus on various ambient and meteorological parameters. In Sec. 2, we present the model and the parameter setup. Section 3 focuses on the sensitivity of $H_2$ contrails to various parameters. Also, we present the updated parametrization in this section. Implications of the simulation results are discussed in Sec. 4, and conclusions are drawn in Sec. 5.

**2   Methods**

In this section, we introduce our model (Sec. 2.1) and describe the numerical setup employed in our simulations (Sec. 2.2). Furthermore, we define the specific quantities necessary for evaluating and analyzing the results (Sec. 2.3).





## 2.1 Model

We perform simulations of $H_2$ contrails using the LES model EULAG-LCM. The base LES code EULAG solves the anelastic

formulation of the momentum and energy equations (Smolarkiewicz and Margolin, 1997; Prusa et al., 2008). The microphysical

module LCM, which incorporates a Lagrangian particle-tracking method, has been coupled to EULAG (Sölch and Kärcher,

2010), resulting in the model version EULAG-LCM. This model version has been used in multiple previous studies to explore

the properties of young contrails during the vortex phase (e.g. Unterstrasser and Sölch, 2010; Unterstrasser, 2014; Unterstrasser

and Stephan, 2020) and their transition into contrail-cirrus (Unterstrasser et al., 2017a, b; Unterstrasser, 2020). The model

results have also been used to develop a parametrization of ice crystal loss and the geometric depth of contrails after vortex

break-up (Unterstrasser, 2016).

In the particle-based framework of LCM, ice crystals are represented by simulation particles. Each simulation particle

represents a certain number of real ice crystals with identical properties, such as size and mass.

In this study, we consider the deposition growth and sublimation of ice crystals and latent heat release to be the most

important microphysical processes and switch off several LCM routines, such as aggregation and radiation.

## 2.2 Numerical setup and set of simulations

Generally, we assume ambient conditions such that contrails form and persist, i.e. the atmosphere is sufficiently cold and moist

(Schumann, 1996; Gierens and Spichtinger, 2000).

As hydrogen propulsion technology is likely to be initially employed on smaller aircraft, we conduct simulations for both

an aircraft with a wing span of $60.3\,\mathrm{m}$ (A350/B777-like aircraft, default) and a smaller aircraft with a wing span of $34.4\,\mathrm{m}$

(A320/B737-like aircraft). Generally, the setups are consistent with a previous EULAG-LCM study, where contrails from six

different aircraft types were systematically investigated (Unterstrasser and Görsch, 2014). Throughout the text, when we use

"A350" ("A320") aircraft, we are specifically referring to an "A350/B777-like" aircraft ("A320/B737-like" aircraft).

The numbers of grid points in our default model domain are $n_x = 384$, $n_y = 200$, and $n_z = 600$, where $x$, $y$, and $z$ denote the

transverse, longitudinal (along flight direction), and vertical direction, respectively. The size of the simulation domain depends,

e.g., on stratification: A weaker stratification requires a larger simulation domain as the vortices descend further downwards.

Moreover, the vortices exhibit a greater meandering because the Crow instability is more pronounced (Crow, 1970). The grid

resolution is $1\,\mathrm{m}$ along the transverse and vertical and $2\,\mathrm{m}$ along the longitudinal direction. For simulating an A320 aircraft, the

resolution is changed to $\mathrm{d}x = \mathrm{d}z = 0.57\,\mathrm{m}$ and $\mathrm{d}y = 1.14\,\mathrm{m}$ as described in Unterstrasser and Görsch (2014). For both aircraft

types, the domain size in flight direction is chosen to allow the formation of the most unstable Crow mode. Periodic boundary

conditions are applied in the horizontal and longitudinal directions, while rigid boundary conditions are used in the vertical

direction. The total simulated time is around six to seven minutes, where the time step increases from initially $0.03\,\mathrm{s}$ to $0.08\,\mathrm{s}$

at later stages of the simulation.

The simulations start at a plume age of several seconds. At this stage, it can be assumed that the ice crystal formation is

finished and the vortices have fully rolled up. We initialize the ice crystals in two discs (one per wing) with uniform ice crystal





number concentrations. The flow field is a superposition of a turbulent background field and two counter-rotating Lamb-Oseen vortices. The vortex circulation, vortex core radius, and plume radius depend on the aircraft type and are given in Tab. 1.

We incorporate the effects of a hydrogen propulsion system by varying the initial number of ice crystals $N_0$ and the initial amount of emitted water vapor $I_0$. The default values of the initial ice crystal number correspond to typical values of the

fuel flow rate, denoted by $\dot{m}_F$ (as reported in Unterstrasser and Görsch (2014)), and an apparent ice emission index $EI_{iceno} = 2.8 \times 10^{14}\,\mathrm{kg}^{-1}$, which represents a typical kerosene contrail. $N_0$ is then determined by

$$N_0 = m_C \times EI_{iceno}, \tag{1}$$

where the fuel consumption $m_C$ is defined as $m_C := \dot{m}_F/U_\infty$. $U_\infty$ represents the aircraft velocity, and $\dot{m}_F$ denotes the fuel flow rate. The reference values are $N_{0,ref} = 0.85 \times 10^{12}\,\mathrm{m}^{-1}$ for an A320 aircraft and $3.38 \times 10^{12}\,\mathrm{m}^{-1}$ for an A350 aircraft.

To explore a broad range of scenarios, we scale $N_0$ up and down by factors of 10 and 100 relative to $N_{0,ref}$. The downscaling simulations are supposed to cover hydrogen combustion scenarios. Conversely, the upscaling simulations could represent scenarios involving a hydrogen fuel cell propulsion system with HDN occurring (see Sec. 1). Clearly, the assumptions on the initial ice crystal number are made for hypothetical aircraft designs, and the research on contrail formation processes in $H_2$ plumes started only recently (Bier et al., 2024; Ponsonby et al., 2024). A more in-depth discussion of the representativity of

our $N_0$ choice is deferred to Sec. 4.

Analogously to $N_0$, the emission of water vapor is calculated using

$$I_0 = m_C \times EI_{H_2O}, \tag{2}$$

where $EI_{H_2O}$ is the emission index of water vapor, with a value of $1.26\,\mathrm{kg\,kg}^{-1}$ for all aircraft types (Unterstrasser and Görsch, 2014). We increase $I_0$ by multiplying the default values, $3.7\,\mathrm{g\,m}^{-1}$ for an A320 aircraft and $15.0\,\mathrm{g\,m}^{-1}$ for an A350 aircraft

(referred to as $I_{0,kero}$), with 2.57 resulting in $9.51\,\mathrm{g\,m}^{-1}$ and $38.55\,\mathrm{g\,m}^{-1}$ (referred to as $I_{0,H_2}$), respectively. The value of 2.57 corresponds to the ratio of the emission index of water vapor divided by the combustion heat $Q$ of hydrogen to kerosene (Bier et al., 2024). Note that $N_0$ and $I_0$ are given in units "per meter of flight path". Table 2 summarizes the key differences between hydrogen and kerosene fuel and exhaust properties.

We want to mention one caveat that is crucial for properly interpreting the comparison between the $H_2$ and kerosene scenar-

ios. According to Tab. 2, switching from kerosene to $H_2$, the water vapor emission index increases by a factor 7.1, while the specific heat of combustion is higher by a factor of 2.79. The energy-specific water emission is then 2.57 times higher, which is, by the way, a crucial difference affecting the contrail formation process and potential contrail coverage (Schumann, 1996; Bier et al., 2024; Kaufmann et al., 2024). Achieving the same work rate with both fuel types, the hydrogen fuel consumption is 2.79 times lower, and the water vapor emission (per flight distance) $I_0$ is 2.57 times larger. Moreover, it is our design choice

to assume the same $N_{0,ref}$ value for both fuel scenarios. This implies that the reference apparent ice emission index $EI_{iceno,ref}$ of the $H_2$ scenarios (i.e. those simulations with higher $I_0$ values) is actually larger by a factor of 2.79. Comparing a factor "100 down" simulation with larger $I_0$ values (as a typical representative of a $H_2$ contrail) with a conventional kerosene contrail (i.e. reference $N_0$ and smaller $I_0$ values), must not be interpreted as a factor-100 variation of $EI_{iceno,ref}$. The difference is actually smaller (factor $\sim 36$).





Following e.g. Unterstrasser and Sölch (2010), the initial ice crystal size distribution is represented by a lognormal distri-
bution with a width parameter $r_{\mathrm{SD}}$. We adopt a default value of $r_{\mathrm{SD}} = 3.0$ and explore variations with a narrower ($r_{\mathrm{SD}} = 1.0$,
monodisperse) and broader ($r_{\mathrm{SD}} = 4.0$) initial size distribution, as previous studies have identified this parameter as significant
(Unterstrasser and Sölch, 2010; Unterstrasser, 2014). Note that other modelling studies that initialize the contrails at an earlier
state without the specification that all emitted water vapor has already been deposited on the ice crystals show a weaker impact
of the width parameter (Lewellen et al., 2014). We conducted simulations only with a 100-fold increase or decrease when
altering the width parameter (to reduce the number of simulations).

Figure 1 shows the mean ice crystal radius versus the initial number of ice crystals for kerosene and hydrogen water vapor
emission. A higher $N_0$ value generally results in a smaller mean radius because the total amount of water is distributed among
more particles, causing each particle to receive less water. It is also evident that the initial water content $I_0$ directly affects the
initial particle size: The higher $I_0$ (indicated by triangles that represent the hydrogen case), the larger the initial particles. The
initial mean ice crystal radii remain unaffected by a variation in the initial width of the size distribution $r_{\mathrm{SD}}$.

Additionally, we investigate the influence of different atmospheric conditions on the evolution of contrail ice crystals. At the
lower boundary of the simulation domain, the pressure is $250\,\mathrm{hPa}$ and the air density is roughly $0.4\,\mathrm{kg\,m^{-3}}$ changing with the
prescribed ambient temperature. We use a background turbulence field with an eddy dissipation rate $\epsilon = 10^{-7}\,\mathrm{m^2\,s^{-3}}$. Previous
research has highlighted the importance of ambient temperature at cruise altitude, ambient relative humidity, and atmospheric
stratification (described by the Brunt-Väisälä frequency) as key input parameters (Lewellen et al., 2014; Unterstrasser, 2016).
We vary these parameters as outlined in Tab. 1. Values that are marked with a star refer to the default simulation of an A350
aircraft with water vapor emission that corresponds to a kerosene combustion system $I_0 = 15.0\,\mathrm{g\,m^{-1}}$ and initial ice crystal
number $N_0 = 3.38 \times 10^{12}\,\mathrm{m^{-1}}$ at ambient temperature of $217\,\mathrm{K}$, ambient relative humidity with respect to ice of $120\,\%$, and
a standard value for atmospheric stability of $1.15 \times 10^{-2}\,\mathrm{s^{-1}}$. Ice crystal formation is improbable for ambient temperatures
exceeding $233\,\mathrm{K}$, as this surpasses the homogeneous freezing temperature (Bier et al., 2024), and liquid droplets that form first
in the cooling exhaust plume would not freeze. Nevertheless, we include simulations with ambient temperatures of $233\,\mathrm{K}$ and
$235\,\mathrm{K}$ as limiting cases, as droplet freezing in turbulent and quickly cooling plumes is not well-constrained. An overview of
all performed simulations is provided in Tab. A1.

## 2.3 Quantities of interest

In the following, we give definitions of quantities that are used throughout the paper.

The total number of ice crystals per meter of flight path and the vertical ice crystal number profile are defined as

$$N_{\mathrm{tot}}(t) = \frac{1}{L_y} \int \int \int N(x,y,z,t)\,\mathrm{d}x\,\mathrm{d}y\,\mathrm{d}z \quad \text{and} \tag{3}$$

$$N_{\mathrm{v}}(z,t) = \frac{1}{L_y} \int \int N(x,y,z,t)\,\mathrm{d}x\,\mathrm{d}y. \tag{4}$$





| Aircraft and vortex parameters | | | Ice crystal parameters | | | |
|---|---|---|---|---|---|---|
| | A320/B737 | A350/B777* | | | A320/B737 | A350/B777 |
| $b_{\mathrm{span}} / \mathrm{m}$ | 34.4 | 60.3 | $N_0 / ([10^{10}, 10^{11}, 10^{12*}, 10^{13}, 10^{14}]\,\mathrm{m}^{-1})$ | | 0.85 | 3.38 |
| $\Gamma_0 / (\mathrm{m}^2\,\mathrm{s}^{-1})$ | 240 | 520 | $I_0 / (10^{-3}\,\mathrm{kg\,m}^{-1})$ | | $[3.7^*, 9.51]$ | $[15.0^*, 38.55]$ |
| $r_{\mathrm{c}} / \mathrm{m}$ | 3.0 | 4.0 | $r_{\mathrm{SD}}$ | | $[1.0, 3.0, 4.0]$ | |
| $r_{\mathrm{plume}} / \mathrm{m}$ | 12 | 20 | | | | |
| Atmospheric conditions | | | Numerical parameters | | | |
| $\rho_{\mathrm{air}} / (\mathrm{kg\,m}^{-3})$ | 0.4 | | | | A320/B737 | A350/B777 |
| $p_{\mathrm{amb}} / (10^2\,\mathrm{Pa})$ | 250 | | $\mathrm{d}x, \mathrm{d}y, \mathrm{d}z / \mathrm{m}$ | | 0.57, 1.14, 0.57 | 1, 2, 1 |
| $T_{\mathrm{amb}} / \mathrm{K}$ | $[217^*, 225, 230, 233, 235]$ | | $n_x, n_y, n_z$ | | 384-768, 200, 600-1112 | |
| $RH_{\mathrm{i,amb}} / \%$ | $[110, 120^*]$ | | $t_{\mathrm{sim}} / \mathrm{s}$ | | $\approx 400$ | |
| $\epsilon / (\mathrm{m}^2\,\mathrm{s}^{-3})$ | $10^{-7}$ | | $\mathrm{d}t / \mathrm{s}$ | | 0.03 - 0.08 | |
| $N_{\mathrm{BV}} / (10^{-2}\,\mathrm{s}^{-1})$ | $[0.5, 1.15^*]$ | | | | | |

**Table 1.** Aircraft/vortex, ice crystal, atmospheric, and numerical parameters. Multiple values in square brackets refer to sensitivity studies, and values with a star refer to default values. $b_{\mathrm{span}}$: wingspan; $\Gamma_0$: circulation; $r_{\mathrm{c}}$: vortex core radius; $r_{\mathrm{plume}}$: plume radius at initialization; $N_0$: initial number of ice crystals; $I_0$: initial amount of emitted water vapor; $r_{\mathrm{SD}}$: width of initial ice crystal size distribution; $\rho_{\mathrm{air}}$: air density at cruise altitude; $p_{\mathrm{amb}}$: ambient pressure; $T_{\mathrm{amb}}$: ambient temperature at cruise altitude; $RH_{\mathrm{i,amb}}$: ambient relative humidity with respect to ice; $\epsilon$: eddy dissipation rate; $N_{\mathrm{BV}}$: Brunt-Väisälä frequency; $\mathrm{d}x, \mathrm{d}y, \mathrm{d}z$: mesh sizes; $n_x, n_y, n_z$: number of grid points; $t_{\mathrm{sim}}$: simulated time; $\mathrm{d}t$: time step.

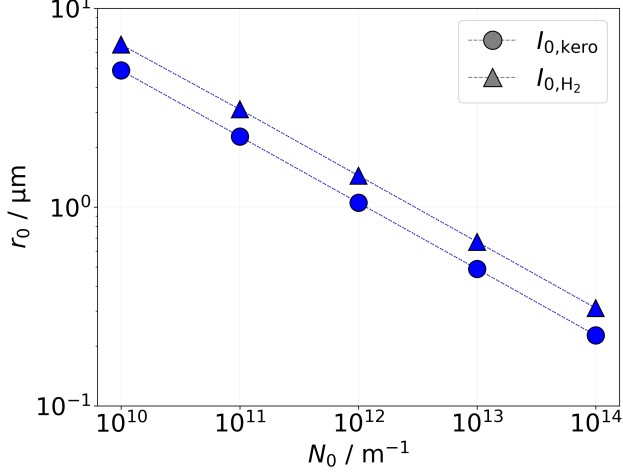

**Figure 1.** Mean initial ice crystal radius as a function of the initial number of ice crystals. Its sensitivity regarding the microphysical initialization in terms of $I_0$ (symbol) is depicted.





| fuel/exhaust parameters | hydrogen | kerosene | ratio |
|---|---|---|---|
| $\mathrm{EI}_{H_2O} / (\mathrm{kg\,kg^{-1}})$ | **8.94** | 1.26 | 7.10 |
| $Q / (10^6\,\mathrm{J\,kg^{-1}})$ | **120** | 43 | 2.79 |
| $\mathrm{EI}_{H_2O}\,Q^{-1} / (10^{-6}\,\mathrm{kg\,J^{-1}})$ | **0.075** | 0.029 | 2.57 |
| | A320/B737 | | |
| $m_C / (10^{-3}\,\mathrm{kg\,m^{-1}})$ | 1.06 | 2.96 | $2.79^{-1}$ |
| $I_0 / (10^{-3}\,\mathrm{kg\,m^{-1}})$ | 9.51 | 3.7 | 2.57 |
| $N_{0,\mathrm{ref}} / (10^{12}\,\mathrm{m^{-1}})$ | 0.85 | 0.85 | 1 |
| $\mathrm{EI}_{\mathrm{iceno,ref}} / (10^{14}\,\mathrm{kg^{-1}})$ | 7.81 | 2.8 | 2.79 |
| | A350/B777 | | |
| $m_C / (10^{-3}\,\mathrm{kg\,m^{-1}})$ | 4.3 | 12.0 | $2.79^{-1}$ |
| $I_0 / (10^{-3}\,\mathrm{kg\,m^{-1}})$ | 38.55 | 15.0 | 2.57 |
| $N_{0,\mathrm{ref}} / (10^{12}\,\mathrm{m^{-1}})$ | 3.38 | 3.38 | 1 |
| $\mathrm{EI}_{\mathrm{iceno,ref}} / (10^{14}\,\mathrm{kg^{-1}})$ | 7.81 | 2.8 | 2.79 |

**Table 2.** Fuel, engine, and exhaust parameters for hydrogen propulsion (second column), kerosene (third column) and the ratio between both (fourth column). The water vapor mass emission index and specific heat of combustion ("lower calorific value") are based on Tab. 1 of Schumann (1996).

Both quantities are averaged along flight direction. The total ice mass and the vertical ice mass profile are computed analogously. The normalized ice crystal number is calculated as

$$f_N(t) = \frac{N_{\mathrm{tot}}(t)}{N_0}. \tag{5}$$

In Sec. 3 we will use the term "survival fraction of ice crystals" $f_{N,s}$, which is the normalized ice crystal number at the end of the simulation. Another pertinent quantity to examine is the number of sublimated ice crystals $N_{\mathrm{subl}}$. Vertical normalized

profiles $f_{N,v,\mathrm{subl}}$ are computed as described in Eqs. 3-5 and track the number of ice crystals that sublimated at a specific altitude.

## 3  Results

### 3.1  Exploring H$_2$ contrails

In this section, we examine the impact of the initial number of ice crystals and the initial amount of water vapor, corresponding to an H$_2$ combustion engine or a potential fuel cell setup. These two parameters are varied as described in Sec. 2.2. The

simulations presented in this section are performed for an A350 aircraft at $T_{\mathrm{amb}} = 217\,\mathrm{K}$, $RH_{i,\mathrm{amb}} = 120\,\%$, and $N_{\mathrm{BV}} = 1.15 \times 10^{-2}\,\mathrm{s^{-1}}$, which were also the baseline meteorological conditions in previous EULAG-LCM studies. Hence, this scenario is





well-explored and the new simulations build upon the existing ones. Unless stated otherwise, the presented results use the
default initial values (indicated by a star in Tab. 1).

Figure 2 (top row) illustrates the temporal evolution of contrail ice crystals in the $x, z$-plane over a six-minute period. The
ice crystal number concentration is averaged along the flight direction. At the start of the simulation, two circular plumes are
initialized at $z = 0\,\mathrm{m}$. Within the first two minutes, a downward motion of the ice crystals is observed as the aircraft exhaust
descends with the downward moving vortices, forming the primary wake. Also, we see a horizontal broadening of the contrail
after few minutes due to the Crow instability (Crow, 1970). Ice crystals that are continuously detrained from the descending
exhaust form a curtain between the original emission altitude and the vortex location, known as the secondary wake (Sussmann
and Gierens, 1999; Unterstrasser, 2014). After six minutes, ice crystals are visible close to or even above the cruise altitude as
the vertically displaced air masses rise back due to buoyancy after vortex break up. Whereas ice crystals in the secondary wake
grow in size due to the deposition of the available water vapor, a significant portion of the ice crystals can sublimate in the
primary wake because of adiabatic heating (Sussmann and Gierens, 1999; Lewellen and Lewellen, 1996; Unterstrasser, 2016).

The second row in Fig. 2 shows the cross-sectional ice crystal number concentrations after six minutes for the reference
$N_0$-case ($N_0 = 3.38 \times 10^{12}\,\mathrm{m}^{-1}$) and the $N_0$-upscaling and $N_0$-downscaling cases. Higher final number concentrations in
absolute terms are evident in the upscaling cases ($N_0 = 3.38 \times 10^{13}\,\mathrm{m}^{-1}$ and $N_0 = 3.38 \times 10^{14}\,\mathrm{m}^{-1}$). Nonetheless, relative to
the initial number, fewer ice crystals survive in these cases, as illustrated in the third row, where the final vertical profiles of
normalized ice crystal numbers (green curves) are depicted. This observation is further supported by the final vertical profiles
of normalized sublimated ice crystals (orange curves). In the upscaling cases, ice crystals begin to sublimate at higher altitudes
because they are smaller in size (see Fig. 1); a phenomenon that has already been described by Huebsch and Lewellen (2006)
and Unterstrasser (2014).

Figures 3(a) and (b) display the temporal evolution of the normalized total ice crystal number and absolute total ice mass for
different $N_0$- and $I_0$-values. The most significant reductions in $f_N$ occur in the upscaling scenarios (red curves in Fig. 3(a)),
while the $N_0$-downscaling simulations (blue curves) show little to no reduction in the temporal evolution of $f_N$. Due to a higher
absolute number of surviving ice crystals in the upscaling cases, these scenarios also exhibit a larger final total ice mass (red
curves lie above the blue ones in Fig. 3(b)). The greater loss of ice crystals in the upscaling scenarios is reflected in the survival
fraction depicted in Fig. 3(c). Conversely, contrails with a lower value of $N_0$ (and thus larger initial crystals) are less prone to
sublimation, resulting in a higher survival fraction. This implies that initial differences in ice crystal numbers diminish over
time. In this specific scenario, initial differences spanning four orders of magnitude (with $N_0$-scaling factors of 0.01 and 100)
reduce to just two orders of magnitude (0.015 and 7.5) after the vortex phase. Moreover, initializing with a greater amount of
emitted water vapor (indicated by circles in Fig. 3(c)) results in larger ice crystal sizes (refer to Fig. 1), which in turn increases
the survival fraction and ice mass. In the $I_{0,\mathrm{H_2}}$ simulations, the reduction in initial differences is less pronounced, with the four
orders of magnitude difference reducing to three orders by the end of the vortex phase.





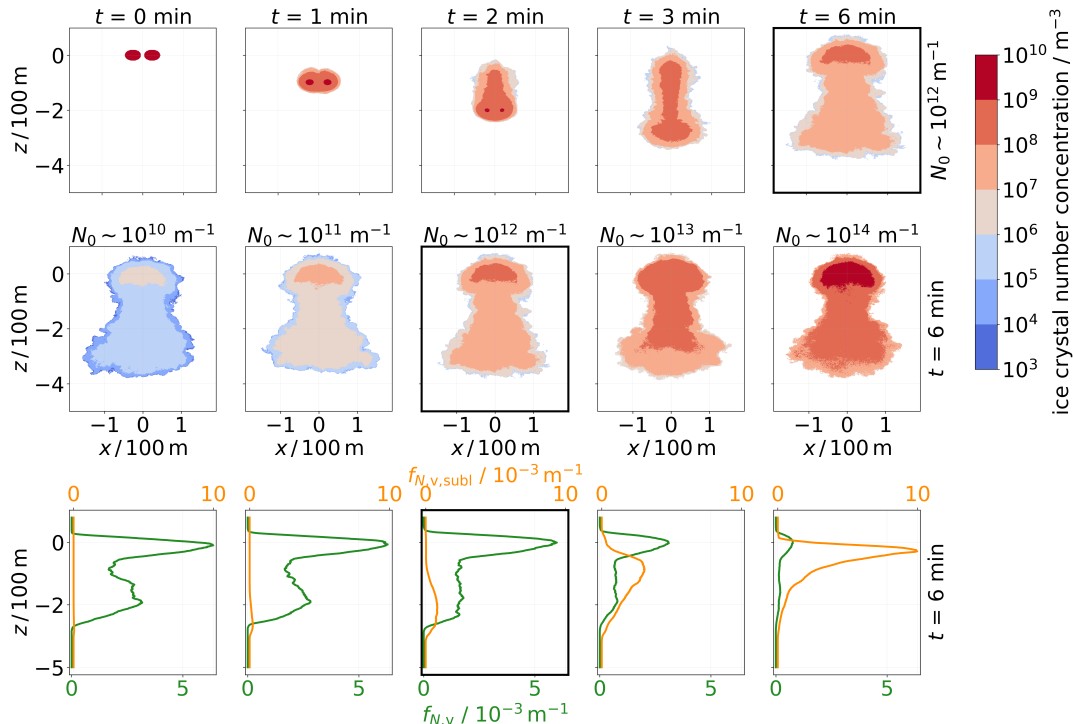

**Figure 2.** Ice crystal number concentrations, averaged along the flight direction in the $x, z$-plane, are depicted in the first and second rows. The first row illustrates the temporal evolution over six minutes. The final concentration distribution is displayed in the second row for five different $N_0$ values. The third row shows the normalized vertical profiles of ice crystals (green) and sublimated ice crystals (orange) (the first row depicts one simulation, and the second and third rows depict five different simulations). The simulation displayed in the top row refers to the simulation in the middle panels in the second and third rows, marked by the black frame. The displayed simulations are performed for an A350 aircraft at $T_\mathrm{amb} = 217\,\mathrm{K}$, $RH_\mathrm{i,amb} = 120\,\%$, and $I_0 = I_{0,\mathrm{H}_2}$.

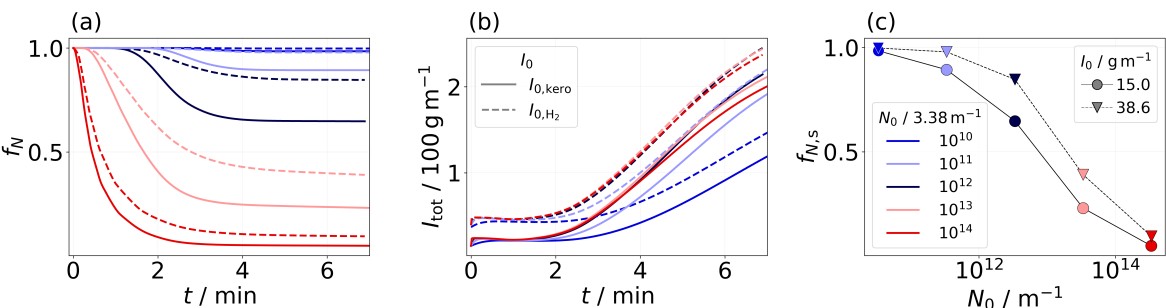

**Figure 3.** Temporal evolution of the normalized total number of ice crystals and total ice mass for different values of $N_0$ and $I_0$ ((a) and (b)). In panel (c), the survival fractions as functions of $N_0$ are depicted.



## 3.2  Parameter study of $H_2$-contrail properties

### 3.2.1  Sensitivity to ambient conditions

As revealed in previous studies, the evolution of contrail ice crystal number and ice mass is highly sensitive to ambient conditions (Lewellen and Lewellen, 2001; Unterstrasser, 2016). This section investigates the impact of ambient relative humidity, temperature, and atmospheric stratification on $H_2$-contrail properties. Note that we performed simulations at $T_{\mathrm{amb}} \geq 230\,\mathrm{K}$ only for $H_2$ contrails with an increased $I_0$ value as the formation of kerosene contrails is unlikely at these high ambient temperatures. The results presented in this section refer to simulations of an A350 aircraft with an initial size distribution width of $r_{\mathrm{SD}} = 3.0$.

In Fig. 4, final vertical profiles of the ice crystal mass and normalized number and survival fractions (after approximately six minutes) are depicted. Columns represent simulations at temperatures of $217\,\mathrm{K}$, $225\,\mathrm{K}$, $230\,\mathrm{K}$, $233\,\mathrm{K}$, and $235\,\mathrm{K}$, respectively. The simulations shown in the first and second rows are performed with $I_0 = I_{0,\mathrm{H}_2}$. The total ice mass (first row (1)) remains largely unaffected by variations in $N_0$, except the factor 100 downscaling simulation, which shows a significantly reduced ice mass. A notable increase in ice mass is observed with increasing temperature, attributable to the larger absolute amount of water vapor in the atmosphere. This additional water vapor can be deposited onto the ice crystals, thereby increasing their size.

When examining the normalized ice crystal number (second row (2)), we observe a reduction with increasing temperature, primarily in the primary wake. At higher ambient temperatures, the excess water vapor emitted by the engines causes a smaller increase in ice supersaturation in the plume due to the nonlinear relationship between saturation vapor pressure and temperature. Also, this leads to a greater decrease in relative humidity within the descending vortices, resulting in an increased sublimation. Hence, ice crystals are more prone to sublimation in the primary wake at higher ambient temperatures (Unterstrasser, 2016; Bier and Burkhardt, 2022). This phenomenon is most pronounced in the $N_0$-upscaling cases.

The sensitivity of the survival fraction to the ambient relative humidity and ambient temperature is illustrated in the bottom row (3). In addition to the default cases with $I_0 = I_{0,\mathrm{H}_2}$, data points for the kerosene reference case with $I_0 = I_{0,\mathrm{kero}}$ are displayed for $T_{\mathrm{amb}} = 217\,\mathrm{K}$ and $225\,\mathrm{K}$ (see yellow symbols). Simulations with $I_0 = I_{0,\mathrm{H}_2}$ indicate that ice crystals, which would typically sublimate under $I_{0,\mathrm{kero}}$ conditions, are now sufficiently large to withstand adiabatic heating. This effect is most pronounced in the $RH_{\mathrm{i,amb}} = 110\,\%$, $N_0$-downscaling scenarios (the differences between the yellow and the colored squares are larger than the differences between the yellow and colored circles at $N_0 \leqslant 10^{13}\,\mathrm{m}^{-1}$). Simulations at a lower ambient relative humidity (squared symbols) show a stronger loss of ice crystals, a trend consistent across the variations regarding $N_0$ and $I_0$. This is due to the stronger sublimation effects in the primary wake in case of a reduced relative humidity value (Unterstrasser et al., 2014; Unterstrasser, 2016). Notably, in the extreme $N_0$-downscaling scenario ($N_0 \sim 10^{10}\,\mathrm{m}^{-1}$) at $120\,\%$ relative humidity, the survival fraction remains largely unaffected by the ambient temperature, with survival fractions equal or close to one at both $217\,\mathrm{K}$ ($f_{N,\mathrm{s}} = 1.00$) and $235\,\mathrm{K}$ ($f_{N,\mathrm{s}} = 0.95$). In contrast, for the extreme $N_0$-upscaling scenario, the impact of ambient temperature is significantly more pronounced, reducing the survival fraction by a factor of 4.5, from $f_{N,\mathrm{s}} = 0.09$ at $217\,\mathrm{K}$ down to $f_{N,\mathrm{s}} = 0.02$ at $235\,\mathrm{K}$. This temperature sensitivity is further amplified in drier atmospheric conditions, where the effect on the survival fraction is more substantial. For the $N_0$-downscaling scenario, the survival fraction decreases by a





factor of 2.24, while in the $N_0$-upscaling case, the reduction exceeds even one order of magnitude. In the upscaling cases with ambient temperatures at or above 230 K, nearly all ice crystals sublimate, with $f_{N,\mathrm{s}} < 0.15$. Initial differences in $N_0$ spanning

four orders of magnitude are reduced to 2.98 (217 K, 120 %) and 2.85 (217 K, 110 %), and further decrease to 2.37 (235 K, 120 %) and 2.14 (235 K, 110 %), highlighting the importance of both parameters in the context of ice crystal loss during the vortex phase.

The third ambient parameter varied in this study is stratification. Following e.g. Unterstrasser et al. (2014), we use a Brunt-Väisälä frequency of $N_{\mathrm{BV}} = 1.15 \times 10^{-2}\,\mathrm{s}^{-1}$ as the default, and a smaller value, $N_{\mathrm{BV}} = 0.5 \times 10^{-2}\,\mathrm{s}^{-1}$, which characterizes

an atmosphere with weaker atmospheric stability. The simulation domain needs to be larger in the latter case as the downward movement and oscillation of the vortex system are stronger. Therefore, to reduce the number of simulations and associated computational costs, we varied $N_0$ only by a factor of 100 (both up and down). This simulation set is performed only for $T_{\mathrm{amb}} = 217\,\mathrm{K}$.

Figure 5 displays vertical profiles of the final ice crystal number (panel (a)) and number of sublimated ice crystals (panel

(b)) for both stratification scenarios (indicated by linestyle). Only simulations with $I_0 = I_{0,\mathrm{H}_2}$ are displayed here. Evidently, in a weakly stratified ambient atmosphere, the wake vortices generally descend further down. Consequently, sublimation extends to lower altitudes (especially in the default and $N_0$-downscaling cases, displayed by the black and blue curves). This is mainly observed in the simulations with default or lower $N_0$ value as the difference in the final contrail height is about 200 m, compared to about 50 m in the $N_0$-upscaling scenario. This reduced difference in the $N_0$-upscaling case is because most ice crystals are

lost at higher altitudes, independently of stratification. As anticipated, the survival fraction is reduced for contrails evolving in a weakly stratified atmosphere. Decreasing the ambient relative humidity exacerbates crystal loss further. Also, lower survival fractions are observed in scenarios with $I_0 = I_{0,\mathrm{kero}}$, as previously noted (not shown). The original four orders of magnitude difference in $N_0$ decreases to 2.69 and 2.46 (default and weakly stratified, 120 %), and 2.46 and 2.31 (110 %) at $I_0 = I_{0,\mathrm{kero}}$, as well as 2.98 and 2.72 (120 %), and 2.85 and 2.66 (110 %) at $I_0 = I_{0,\mathrm{H}_2}$.

### 3.2.2 Sensitivity to the microphysical initialization

We prescribe the initial ice crystal size distribution (SD) as a lognormal distribution with a specified width parameter $r_{\mathrm{SD}}$, see e.g. Unterstrasser and Sölch (2010). A variation of $r_{\mathrm{SD}}$ has been shown to have a non-negligible effect on the contrail's evolution during the vortex phase (Unterstrasser, 2014). Varying the width, we adjust the geometric mean diameter such that the total initial ice mass and ice crystal number are unaffected (Unterstrasser, 2014).

In Fig. 6, we present simulations, where $r_{\mathrm{SD}}$ is varied according to Tab. 1. The final contrail height for each $N_0$ scenario appears relatively unaffected by variations in $r_{\mathrm{SD}}$, as shown in Fig. 6(a). However, a narrower initial SD leads to a higher number of ice crystals present at lower altitudes (dashed curves). This can be explained by the delayed onset of sublimation, which occurs at lower altitudes (see panel (b)). In contrast, a broader initial SD results in a higher proportion of smaller ice crystals at the start, which tend to sublimate at higher altitudes, leading to lower survival fractions. This pattern is observed consistently

across all $N_0$ scenarios. To assess the robustness of these results with respect to ambient temperature, we repeated this set

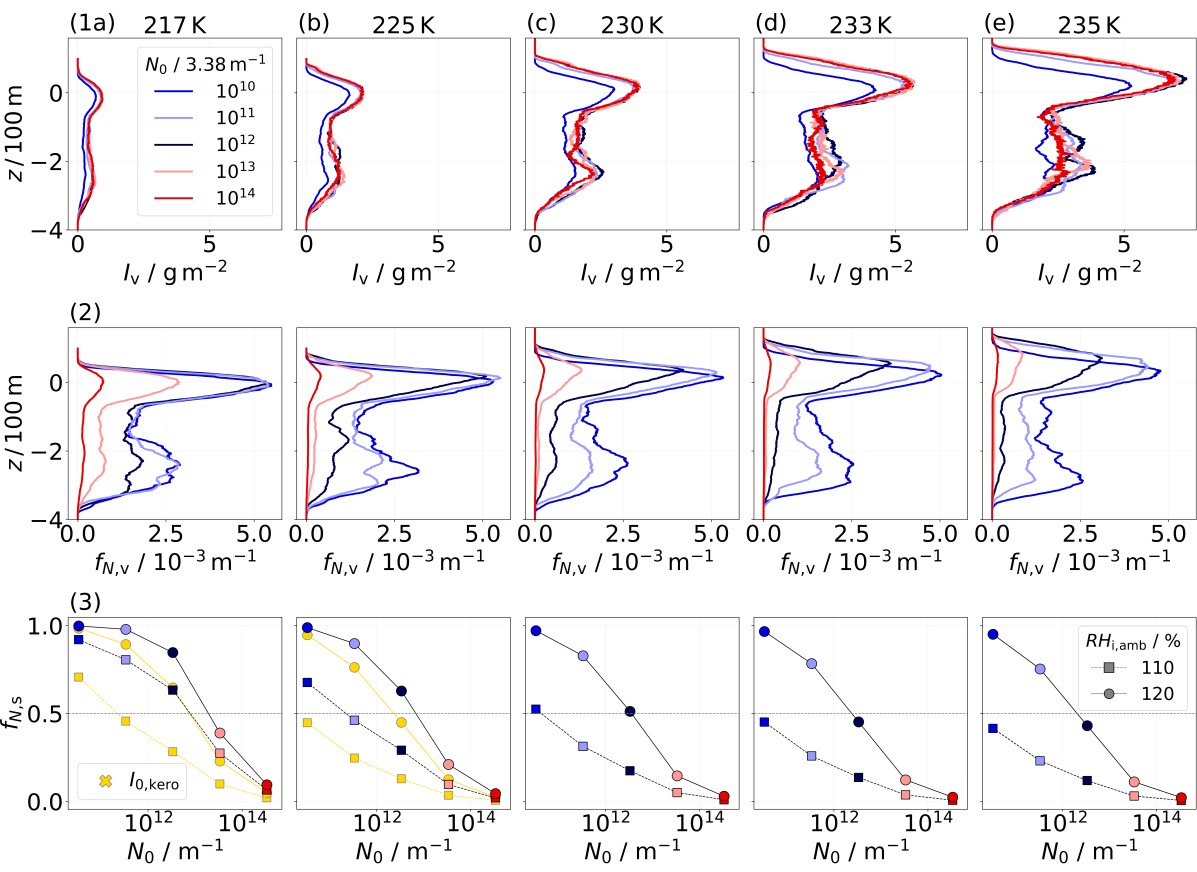

**Figure 4.** Final vertical profiles of ice mass (first row) and normalized ice crystal number (second row) for different $N_0$ values (as given in the legend in panel (1a)) after six minutes. Columns depict results for ambient temperatures of 217 K, 225 K, 230 K, 233 K, and 235 K (see titles on top). The ambient relative humidity is 120 %. The third row displays the survival fraction at the two different humidity values as indicated in panel (3e). At $T_{amb} = 217$ K and 225 K, yellow symbols, which refer to simulations with $I_0 = I_{0,kero}$, are plotted for comparison. A horizontal dashed line at $f_{N,s} = 0.5$ is provided as a visual guide.

1low["





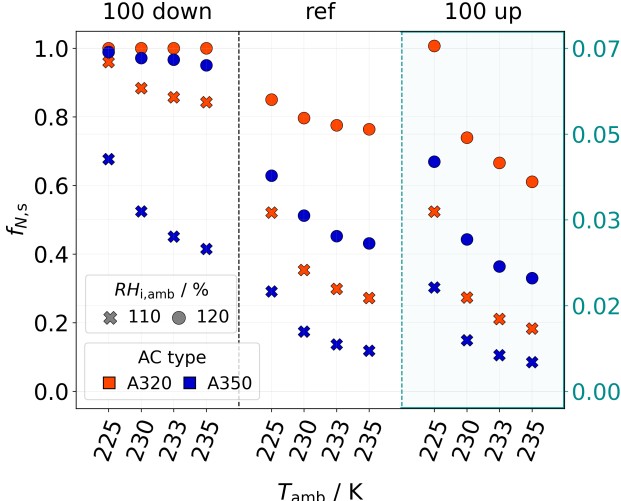

**Figure 7.** Survival fractions of an A320 (red) and an A350 (blue) aircraft. The three boxes represent the ice crystal number scaling as indicated in the title, where, e.g. "100 down" refers to the simulation where $N_0$ is $3.38 \times 10^{10}\,\mathrm{m}^{-1}$ (A350) or $0.85 \times 10^{10}\,\mathrm{m}^{-1}$ (A320). In the case of a factor 100 upscaling, the survival fractions are small. Hence, we adapted the axis as indicated by the light blue labels. Simulation results for both values of ambient relative humidity are shown (symbol).

of simulations at $T_{\mathrm{amb}} = 233\,\mathrm{K}$. The general trends related to $r_{\mathrm{SD}}$ remain unchanged by the variation in ambient temperature. However, as expected based on previous findings, we observe lower survival fractions at this temperature.

In aircraft plumes characterized by negligible soot emission, such as those from hydrogen-powered engines, ice nucleation is assumed to predominantly occur on entrained ambient aerosol particles (Bier et al., 2024). The dry radii of these ambient aerosols, therefore, govern the size of the resulting ice crystals. Bier et al. (2024) identified several aerosol modes with sizes spanning from few nanometers to several micrometers. This broad size range is more accurately captured by an initial size distribution with larger values of $r_{\mathrm{SD}}$ (e.g. 3.0 and 4.0), thus considered more representative of the physical conditions.

### 3.2.3 Sensitivity to aircraft type

H$_2$-contrail simulations for a smaller A320aircraft with wing span of $b_{\mathrm{span}} = 34.4\,\mathrm{m}$ are performed for $T_{\mathrm{amb}} \geq 225\,\mathrm{K}$. The grid spacing is adapted as outlined in Tab. 1. We simulate this aircraft type only with a $N_0$-scaling factor of 100.

Unterstrasser and Görsch (2014) have done baseline simulations for an A320 aircraft in typical kerosene conditions. As revealed in this study, the altitude where sublimation starts is independent of the aircraft type. As the vortex system of the A320 descends more slowly, the sublimation threshold altitude is reached later. The vortices of the A320 dissolve before such low altitudes can be reached, where strong sublimation occurs. This decreases sublimation effects in the primary wake, allowing more ice crystals to survive the vortex phase for smaller aircraft like the A320 than larger ones like the A350. This



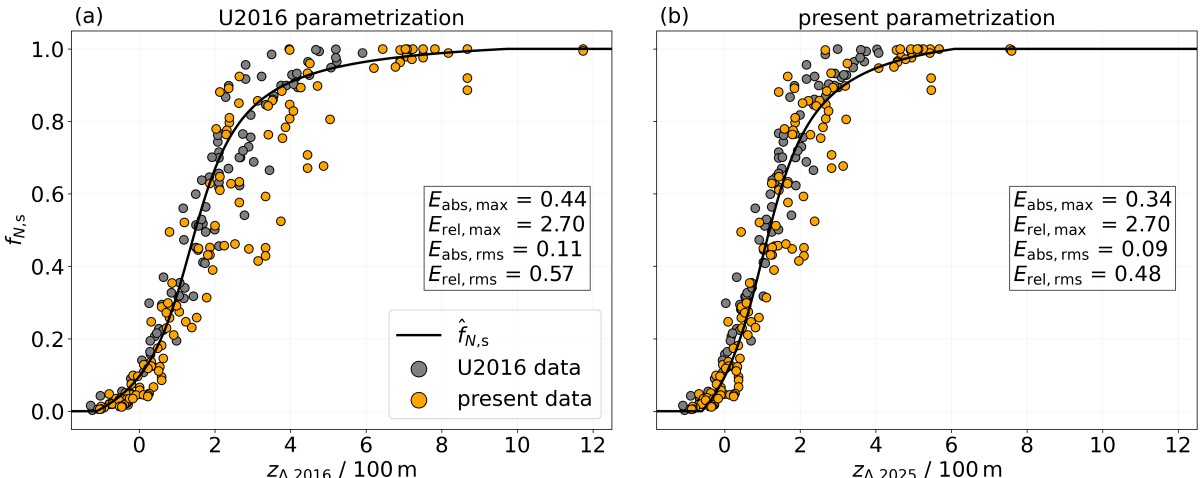

**Figure 8.** Simulated survival fraction as a function of the parameter $z_\Delta$. The original (a) and updated (b) parametrizations are denoted by the solid black line. Grey points represent the simulated survival fractions from the 2016 study, while orange points correspond to those from the present study. Since the fit coefficients are adapted in the new formulation, the $z_\Delta$ parameter differs in both versions, denoted by $z_{\Delta,2016}$ and $z_{\Delta,2025}$.

trend holds across all $N_0$-scaling scenarios. Although the final A350 contrail still contains more ice crystals than the A320 contrail, the relative ice crystal loss is greater for the A350, reducing the initial large differences in ice crystal numbers.

As described in Sec. 2.2, both $N_0$ and $I_0$ are linearly dependent on the fuel flow rate. Since $EI_{iceno}$ and $EI_{H_2O}$ are assumed to be constant across all aircraft types, the mean mass and radius of the initial ice crystals are consequently independent of the
aircraft type, as both are proportional to the ratio $I_0/N_0$. Despite growing larger in the first few seconds in the A350 case due to the larger water vapor emission, more ice crystals are lost within the descending vortices. The stronger vertical displacement of the primary wake and the associated sublimation there outweigh the initial higher ice crystal growth rate, leading to a more significant overall ice crystal loss.

Figure 7 shows how the survival fractions depend on the aircraft type. Each box, delineated by the dashed vertical lines,
corresponds to a different $N_0$-scaling scenario. Note that in the factor 100 upscaling scenario, where the survival fractions are less than 0.1, the $y$-axis is adjusted accordingly. For each scenario, simulations at $T_{amb} = 225\,\text{K}$, $230\,\text{K}$, $233\,\text{K}$, and $235\,\text{K}$ are performed, as indicated on the $x$-axis. As discussed in Sec. 3.2.1, high ambient temperatures and low relative humidity increase ice crystal loss. Overall, the A320 exhibits higher survival fractions, with the most pronounced differences at low $N_0$ values and low ambient humidity (e.g. $f_{N,s} = 0.84$ for the A320 and $f_{N,s} = 0.42$ for the A350 at $T_{amb} = 235\,\text{K}$). In scenarios with a
strong $N_0$ upscaling, nearly all ice crystals sublimate regardless of aircraft type ($f_{N,s} \leq 0.07$).





### 3.3 Parametrization of ice crystal loss

Unterstrasser (2016, from now on U2016) presents an ice crystal loss parametrization that approximates the survival fraction of ice crystals after the vortex phase $f_{N,\mathrm{s}}$ and takes into account the impact of ambient relative humidity, temperature at cruise altitude, thermal stratification, apparent ice crystal emission index, and aircraft parameters. Relevant aircraft parameters are the water vapor emission $I_0$, the wing span $b_{\mathrm{span}}$, and the wake vortices' initial circulation $\Gamma_0$. A typical application of the parametrization is the implementation in larger scale contrail models where the contrail initialization refers to some state after vortex break-up, and effects of the wake dynamics cannot be explicitly resolved. The development of the parametrization was based on a database of $> 80$ EULAG-LCM simulations, which are sensitivity studies with variations of the parameters above. Notably, the focus at that time was on contrails from kerosene combustion. The new set of $H_2$-contrail simulations investigates the sensitivity to $N_0$ more systematically and over a broader range. Moreover, contrails at temperatures above $225\,\mathrm{K}$ have not been considered previously, and the validity of the parametrization beyond this temperature threshold is not guaranteed.

The following paragraphs will briefly repeat the parametrization formulation, describe appropriate adaptations, and analyze its applicability and performance for the new $H_2$-contrail data set.

The ice crystal loss during the vortex phase can be assessed by introducing three length scales that describe the processes relevant to downward propagating contrail ice crystals.

- $z_{\mathrm{desc}}$ is the final vertical displacement of the wake vortex system, which leads to a maximum adiabatic heating experienced in the primary wake. The quantity depends most strongly on thermal stratification, aircraft mass, and wingspan.

- $z_{\mathrm{atm}}$ measures the effect of the ambient supersaturation on the ice crystal mass budget. $z_{\mathrm{atm}}$ is the distance an air parcel has to travel down until its saturation pressure equals its vapor pressure (i.e. until its supersaturation is depleted due to adiabatic heating). The quantity depends most strongly on the ambient relative humidity.

- $z_{\mathrm{emit}}$ measures the effect of the water vapor emission on the ice crystal mass budget. Analogously to $z_{\mathrm{atm}}$, $z_{\mathrm{emit}}$ corresponds to an adiabatic heating such that an initially saturated parcel remains at saturation when the emitted water vapor is added to the parcel. The quantity depends on the ambient temperature and the amount of emitted water vapor, which in turn depends on the fuel consumption, the emission index of water vapor, and the contrail cross-section.

The exact definitions of these length scales are given in Sec. 3.1 of U2016. However, we implement slightly redefined formulae. In the previous version, the definitions of $z_{\mathrm{atm}}$ and $z_{\mathrm{emit}}$ are based on the assumption of balancing water vapor concentrations. In the updated parametrization, we redefine the two length scales by requiring the conservation of water vapor mass mixing ratios, which are the quantities that are actually conserved under adiabatic changes. Hence, we introduce the adiabatic index $\kappa = 3.5$, and the modified equations read as

$$(1 + s_{\mathrm{i}})\, \frac{e_{\mathrm{s}}(T_{\mathrm{CA}})}{T_{\mathrm{CA}}{}^{\kappa}} = \frac{e_{\mathrm{s}}(T_{\mathrm{CA}} + \Gamma_{\mathrm{d}}\, z_{\mathrm{atm}})}{(T_{\mathrm{CA}} + \Gamma_{\mathrm{d}}\, z_{\mathrm{atm}})^{\kappa}} \tag{6}$$

and

$$\frac{e_{\mathrm{s}}(T_{\mathrm{CA}})}{R_{\mathrm{WV}}\, T_{\mathrm{CA}}{}^{\kappa}} + \frac{\rho_{\mathrm{emit}}}{T_{\mathrm{CA}}{}^{\kappa-1}} = \frac{e_{\mathrm{s}}(T_{\mathrm{CA}} + \Gamma_{\mathrm{d}}\, z_{\mathrm{emit}})}{R_{\mathrm{WV}}\, (T_{\mathrm{CA}} + \Gamma_{\mathrm{d}}\, z_{\mathrm{emit}})^{\kappa}}. \tag{7}$$



$\Gamma_d$ is the dry adiabatic lapse rate, with a value of $9.8\,\mathrm{K\,km^{-1}}$. According to U2016, we then define a linear combination of these length scales

$$z_\Delta = \tilde{\alpha}_{\mathrm{atm}}\, z_{\mathrm{atm}} + \tilde{\alpha}_{\mathrm{emit}}\, z_{\mathrm{emit}} - \tilde{\alpha}_{\mathrm{desc}}\, \hat{z}_{\mathrm{desc}} \tag{8}$$

with positive weights $\tilde{\alpha}_{\mathrm{X}}$.

$z_\Delta$ is large if the buffer effect of the ambient supersaturation and the emitted water vapor outweighs the adiabatic heating due to the wake vortex descent. This means, the water vapor surplus is sufficient to keep the heated air parcel supersaturated. [*Side remark: Note that in the simulations, the air parcel does not remain supersaturated, but $RH_i$ quickly relaxes to 100 %. Figure 3b reveals that the water vapor surplus deposits on the ice crystals within a few seconds (for large $N_0$) or within at most 30 s (for low $N_0$). The local maximum attained within the first half a minute clearly relates to the water vapor surplus as estimated by the two length scales $z_{atm}$ and $z_{emit}$. Note that the increase in ice mass after two minutes is due to detrained ice crystals outside of the vortex system, which grow in the supersaturated ambient air. Figure 2a in Unterstrasser and Sölch (2010) displays the evolution of the total ice mass in the descending wake vortex system. This nicely reveals the dependence of the peak value on the water vapor surplus and the subsequent monotonic decrease of the ice mass. Hence, the buffer effect is achieved by increasing the total mass and the mass concentrations of the ice crystals in the vortex system. The larger those quantities are after the initial growth period, the more and the longer the ice crystals can shrink until complete sublimation.*]

A small $z_\Delta$ (including also negative values) means that the wake vortex descent prevails for so long that a substantial fraction or even all the entrained ice crystals are likely lost.

Following U2016, both $\tilde{\alpha}_{\mathrm{atm}}$ and $\tilde{\alpha}_{\mathrm{emit}}$ are modified to include the effect of a variation of $N_0$, while $\tilde{\alpha}_{\mathrm{desc}}$ can be left unchanged (i.e. $\alpha_{\mathrm{desc}} = \tilde{\alpha}_{\mathrm{desc}}$):

$$z_\Delta = \Psi^\gamma (\alpha_{\mathrm{atm}}\, z_{\mathrm{atm}} + \alpha_{\mathrm{emit}}\, z_{\mathrm{emit}}) - \alpha_{\mathrm{desc}}\, \hat{z}_{\mathrm{desc}}. \tag{9}$$

In U2016, $\Psi$ was given by $\frac{1}{\mathrm{EI}^*_{\mathrm{iceno}}}$ with $\mathrm{EI}^*_{\mathrm{iceno}} = \frac{\mathrm{EI}_{\mathrm{iceno}}}{\mathrm{EI}_{\mathrm{iceno,ref}}}$ and a positive constant $\gamma$. In the new formulation, $\Psi$ is defined as

$$\Psi = 1/n_0^*, \tag{10}$$

where $n_0$ is an (intermediate) ice crystal number concentration and the starred quantity is the value normalized by a reference value $n_{0,\mathrm{ref}}$. In both approaches, increasing $\mathrm{EI}_{\mathrm{iceno}}$ (in the original formulation) or $n_0$ (in the new formulation) gives smaller weights $\tilde{\alpha}_{\mathrm{atm}}$ and $\tilde{\alpha}_{\mathrm{emit}}$. This reduces the buffer effect and, consequently, $z_\Delta$. This reflects the fact that ice crystal loss becomes more substantial when the water vapor surplus is distributed over more ice crystals with smaller masses on average. With smaller mean ice crystal sizes, a specific fraction of lost ice mass translates into a larger relative fraction of lost ice crystals.

The definition of $n_0$ is deferred to the Appendix (Eq. (A1)) and the motivation for the switch from $\mathrm{EI}^*_{\mathrm{iceno}}$ to $n_0^*$ is explained in Sec. 4.

The three length scales depend only on input parameters and can be evaluated for given meteorological and aircraft properties. Notably, they are independent of $N_0$, and the sensitivity of $z_\Delta$ to $N_0$ enters the parametrization solely by an adaptation of the weights $\tilde{\alpha}_{\mathrm{atm}}$ and $\tilde{\alpha}_{\mathrm{emit}}$.




Plotting the simulated $f_{N,s}$ values as functions of $z_\Delta$ with suitably chosen weights, the data points can be reasonably well approximated by an arc tangent function. Hence, the parametrized survival fraction $\hat{f}_{N,s}$ can then be defined as

$$\hat{f}_{N,s} = \hat{a}(z_\Delta), \tag{11}$$

with

$$\hat{a}(x) = \beta_0 + \frac{\beta_1}{\pi}\arctan\left(\alpha_0 + (x/100\,\mathrm{m})\right). \tag{12}$$

In the following, we outline our procedure for determining the new set of fit coefficients. Simulations with $r_{SD} = 1.0$ and $r_{SD} = 4.0$, accounting for approximately 25 % of the total dataset, were excluded from the fitting process, as they represent extreme cases of the size distribution width. However, these simulations are still shown in Fig. 8. We selected a representative subset of simulations using the default value of $r_{SD} = 3.0$. Additionally, some data points from simulations with weaker stability were identified as outliers and excluded. Consequently, 69 % of the new simulations and the entire data set from U2016 were used to derive the fit coefficients. A weighting was applied in the fitting procedure, with all data points at $RH_{i,amb} = 110\,\%$ and $EI^*_{iceno} > 1$ being assigned higher weights (1.1 and 2.0, respectively). On the one hand, these weights were introduced to reduce the absolute errors of the $RH_{i,amb} = 110\,\%$ cases. This is favorable as the survival fractions are systematically smaller than for the $RH_{i,amb} = 120\,\%$ cases. Hence, similar absolute errors over both sets of simulations would imply larger relative errors for $RH_{i,amb} = 110\,\%$ cases. Furthermore, the occurrence frequency of ice supersaturation is exponentially distributed with decreasing encounter probabilities for increasing $RH_{i,amb}$ values (Petzold et al., 2017). Increasing the weights is a pragmatic solution to put more emphasis on the low-$RH_{i,amb}$ cases.

We propose the following values for the fitting coefficients:

$$\beta_0 = 0.42, \tag{13a}$$
$$\beta_1 = 1.31, \tag{13b}$$
$$\alpha_0 = -1.00, \tag{13c}$$
$$\alpha_{atm} = 1.27, \tag{13d}$$
$$\alpha_{emit} = 0.42, \tag{13e}$$
$$\alpha_{desc} = 0.49, \tag{13f}$$
$$\gamma = 0.16. \tag{13g}$$

Figure 8 presents a comparison between both versions, where the simulation data used to derive the original parametrization formulation are displayed in grey, and the present (additional) data points are shown in orange. Panel (a) depicts the 2016 parametrization curve. The new parametrization (panel (b)) reduces both the absolute maximum and root mean square error (see values inserted in the figure). The absolute root mean square is reduced by 2 %, whereas the relative root mean square error of the U2016 data is decreased by 9 %. Relative to the U2016 database, the new $H_2$ simulations sample particularly the




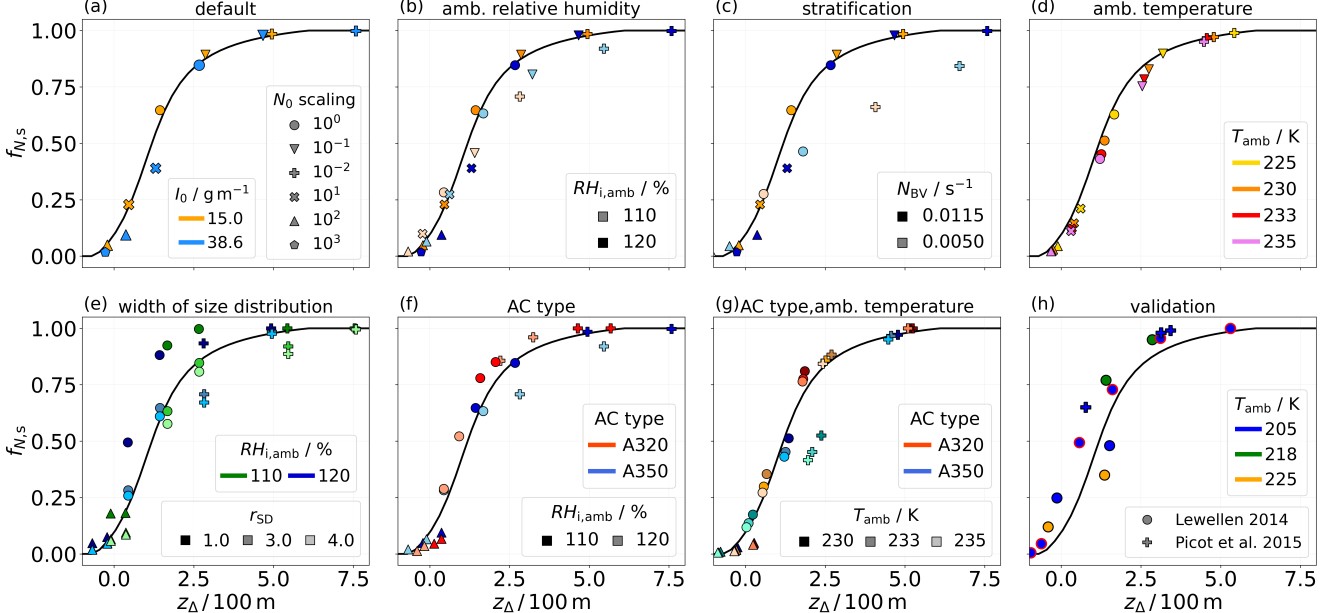

**Figure 9.** Simulated survival fractions as functions of the parameter $z_\Delta$ ($= z_{\Delta,2025}$), focusing exclusively on new $H_2$ simulations. The symbol legend in panel (a) applies to the panels (a)-(g), with symbols indicating $N_0$-scaling information. In panels (a)-(c), blue symbols denote hydrogen water vapor emissions, while orange symbols represent kerosene cases. Each panel highlights a different meteorological or aircraft scenario, as the titles indicate. Panel (h) displays survival fractions from other studies for comparison and validation. There, symbols with a red outline correspond to a study investigating the effects of varying the ice crystal emission index (Lewellen et al., 2014).

space for large and very small $f_{N,s}$ values more densely. The supplement contains further plots that help to rate the changes implied by the new version.

In Fig. 9, we show the updated parametrization for various meteorological and aircraft scenarios. As indicated by its title, each panel highlights specific sensitivity analyses. The dependencies of the survival rate on the input parameters have been thoroughly explained in the preceding sections. The following will discuss the new insights gained from the updated parametrization. It can be observed that simulations performed at temperatures above 225 K are well represented by the parametrization. This holds for both the $N_0$-upscaling and -downscaling scenarios. While a critical temperature exists above which contrail formation is not possible, this temperature threshold has no direct relevance for the microphysical processes during the vortex phase. Hence, it is not surprising that the validity of the parametrization extends beyond 225 K.

The parametrization does not account for variations in $r_{SD}$, meaning that data points from simulations with different $r_{SD}$ values but the same $z_\Delta$ value are treated identically. Notably, for $N_0$-upscaling cases, simulations with $r_{SD} = 1.0$ tend to show survival fractions that are somewhat overestimated. For the time being, this is an irreducible uncertainty partly due to idealizations in our initialization.



Regarding aircraft type, the parametrization effectively represents both A350 (bluish) and A320 (reddish) cases across upscaling and downscaling scenarios, as shown in panels (f) and (g). Additionally, the parametrization is consistent with other simulation results from studies such as Lewellen et al. (2014) and Picot et al. (2015), as highlighted in panel (h). Our updated $H_2$ parametrization not only aligns well with these results, but also particularly captures the ice crystal emission index variation study from Lewellen (2014), further demonstrating the validity of the new parametrization.

### 3.4 Impact of vortex phase processes on contrail-cirrus evolution

This section presents results of contrail-cirrus simulations and demonstrates the relevance of the ice crystal number in young contrails for the subsequent transition into contrail-cirrus.

We employ the EULAG-LCM model in a setup largely based on that used in Unterstrasser et al. (2017a) and Unterstrasser et al. (2017b). We use a domain with two dimensions, $x$ (horizontal) and $z$ (vertical), that is perpendicular to the direction of flight. We embed the data of the vortex phase simulation (averaged along the flight direction) in a significantly larger simulation domain spanning $40\,\mathrm{km}$ in $x$- and $2.5\,\mathrm{km}$ in $z$-direction, with grid resolutions of $\mathrm{d}x = \mathrm{d}z = 10\,\mathrm{m}$. The simulated time is $8\,\mathrm{h}$. While sedimentation was switched off in the vortex phase simulation, it is switched on here as it is a crucial process in the contrail-cirrus transition. The sedimentation velocities are determined using the Reynolds number and the maximum particle dimensions, as described in the Appendix of Sölch and Kärcher (2010). We prescribe a stepwise vertical profile of relative humidity, assuming an ambient ice supersaturated layer of $120\,\%$ between 1 and $2\,\mathrm{km}$ (see further details in, e.g., Unterstrasser et al. (2017a)). We also impose a synoptic-scale updraught (with an updraught velocity $w = 1\,\mathrm{cm\,s^{-1}}$) over roughly six hours, leading to a final adiabatic cooling of $2\,\mathrm{K}$. We choose an A350 aircraft with $T_{\mathrm{amb}} = 225\,\mathrm{K}$, $N_{\mathrm{BV}} = 1.15 \times 10^{-2}\,\mathrm{s^{-1}}$, $I_0 = I_{0,\mathrm{H_2}}$, and $r_{\mathrm{SD}} = 3.0$. In the first simulation series, the contrail-cirrus initialization is directly based on the results from the according vortex phase simulation. The initialized contrail consists of $N = f_{N,\mathrm{s}} \times N_0$ ice crystals. In the second simulation series, we deliberately disregard vortex phase loss processes and initialize a contrail with $N_0$ ice crystals. This is accomplished by using the contrail data from the first simulation series and uniformly scaling up the ice crystal number concentrations by a factor of $(f_{N,\mathrm{s}})^{-1}$.

An important quantity for analyzing contrail evolution is the total extinction, which serves as a metric for the radiative effect of an individual contrail (Unterstrasser, 2020). It is defined as the horizontal integral of the extinction, $1 - \exp(-\tau)$, where $\tau$ represents the optical thickness along the vertical direction. Mathematically, it is expressed as:

$$
\begin{aligned}
E(t) &= \int \left(1 - \exp\left(-\tau(x,t)\right)\right)\mathrm{d}x \approx \int \tau(x,t)\,\mathrm{d}x \\
&= \iint \chi(x,z,t)\,\mathrm{d}z\,\mathrm{d}x,
\end{aligned}
\tag{14}
$$

where $\chi(x,z,t)$ is the extinction coefficient. It is proportional to the projected area of the ice crystals per unit volume. We assume an extinction efficiency $Q_{\mathrm{ext}} = 2$, as described in Schumann et al. (2011). Figure 10 illustrates the temporal evolution of the total ice crystal number (panel (a)) and the total extinction (panel (b)). In both panels, the color represents the $N_0$ scaling, with red indicating upscaling scenarios and blue denoting downscaling scenarios. The line style differentiates between the two setups.





Three key observations emerge: Firstly, $E$ is significantly higher in the $N_0$-upscaled scenarios. This outcome is expected, as total extinction is proportional to the effective surface area of the ice crystals; a larger number of ice crystals, as shown in panel (a), leads to a greater extinction. Secondly, the contrail(-cirrus) dissipates more quickly when $N_0$ is smaller. In the scenario where $N_0$ is reduced by a factor of 100 (dark blue), the contrail disappears after only few hours, indicating a significantly shorter lifetime. Thirdly, we observe an increase in $E$ in the scenarios where the vortex phase losses are disregarded. In these

cases, the contrail is initialized with more ice crystals compared to the first simulation series (see panel (a)).

A qualitative comparison of the two setups can be made by examining the lifetime-integrated total extinction values, $\hat{E} = \int E(t)\,\mathrm{d}t$, that are displayed in Fig. 11(a). Ignoring vortex phase losses leads, at maximum, to a difference in lifetime-integrated total extinction of $46\,\%$. This indicates that the radiative impact of an individual contrail can be significantly overestimated if vortex phase processes, such as contrail expansion and ice crystal sublimation in the primary wake, are not considered. The

non-linear increase in the $\hat{E}$ difference between both setups in the $N_0$-upscaled simulations may be attributed to saturation effects. As the number of ice crystals increases, competition for available water vapor in the spreading contrail-cirrus becomes more important. Consequently, the peak ice mass does not continue to grow indefinitely with increasing $N_0$ values but is expected to reach a limit eventually. This explains why the simulation with a 100-fold increase in $N_0$ shows a smaller increase in lifetime-integrated total extinction when disregarding the vortex phase losses compared to the simulation with a 10-fold

increase in $N_0$. We repeated this kind of simulation series, i.e. disregarding the crystal losses from the vortex phase, for the $I_{0,\mathrm{kero}}$ scenario. Interestingly, we observe only a small impact on the lifetime-integrated total extinction when the vortex phase losses are ignored (see Fig. 11(b)). This suggests that the initial water vapor emission, compared to the ice crystal number, has only a minor impact on the radiative effect of a single contrail-cirrus.

This is just one example of a contrail-cirrus simulation set. A comprehensive analysis, examining various meteorological

parameters, will follow.

## 4    Discussion

This study highlights the importance of ice crystal loss during the vortex phase depending on the propulsion system. While previous research has examined the effect of initial ice crystal numbers on crystal loss — such as Lewellen (2014), who varied the ice crystal emission index across six orders of magnitude, and Unterstrasser (2014), who explored a variation over

two orders of magnitude — this work takes a systematic approach tailored explicitly to $H_2$ contrails. It uniquely combines variations in the initial number of ice crystals with adjustments to the amount of emitted water vapor. The $N_0$-upscaling scenarios may not only represent a potential $H_2$ fuel cell setup but also account for the formation of contrail ice crystals on ultrafine volatile particles (UFPs). UFPs, typically $\mathrm{nm}$-sized particles, can contribute to ice crystal formation if the plume supersaturation is high enough to overcome the Kelvin barrier for activation (Kärcher et al., 2015). This requires the reduction

of the soot particle number at least 10-fold, preventing them from acting as the primary condensation nuclei for water vapor, and an ambient temperature significantly below the formation threshold (Kärcher, 2018).





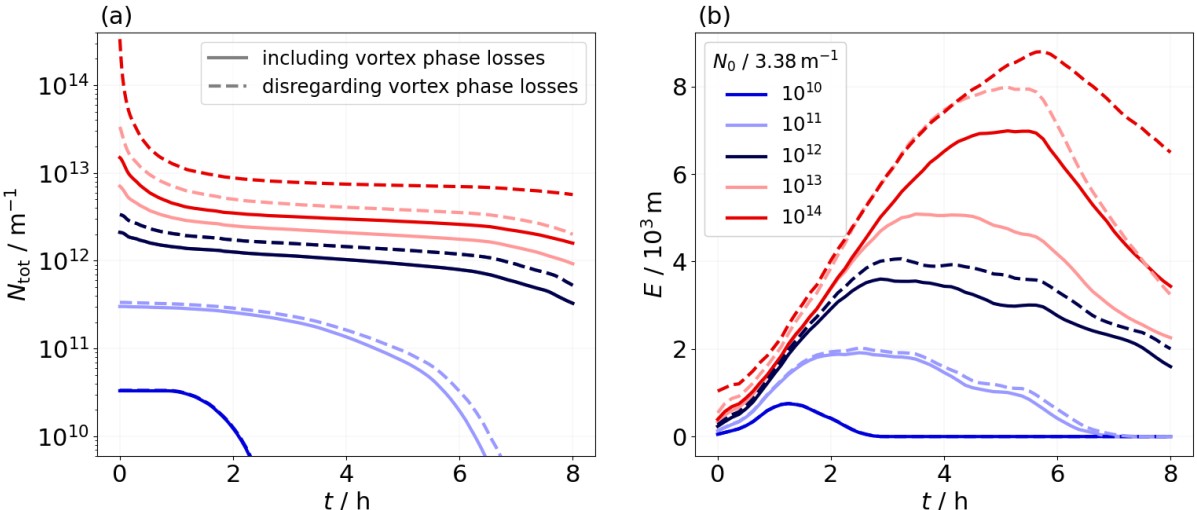

**Figure 10.** Time evolution of the total number of ice crystals (a) and total extinction (b) for the $N_0$ up- and downscaling cases. In the first simulation series, the contrails are initialized by using the final state from the respective vortex phase simulations; in the second simulation series, the ice crystal numbers are upscaled by the respective survival fractions.

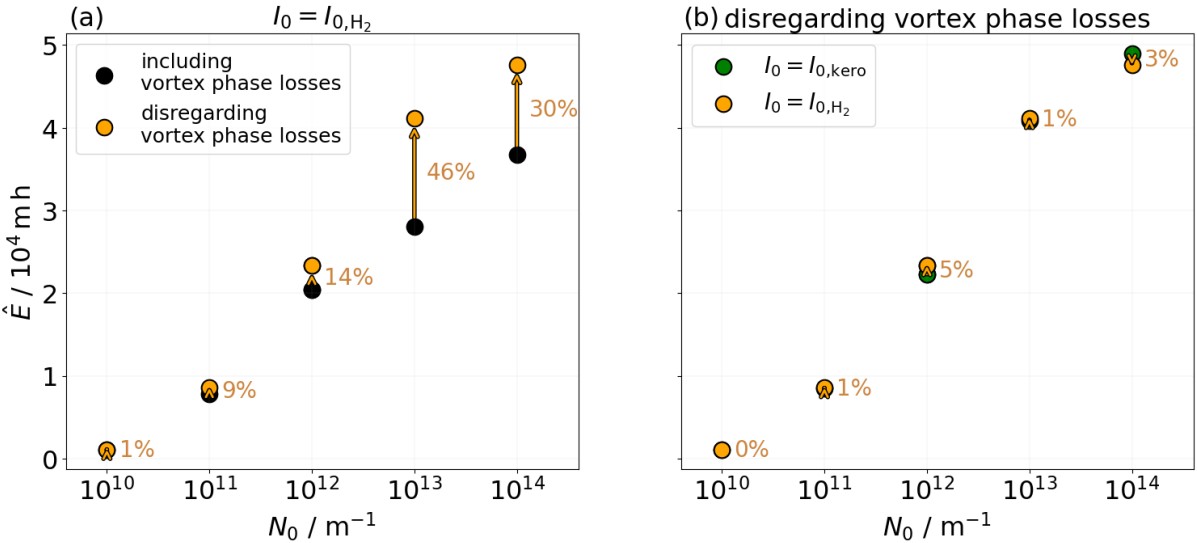

**Figure 11.** Lifetime-integrated total extinction $\hat{E}$ for all five $N_0$-scaling scenarios. Panel (a) highlights the increase of this quantity in the second simulation series (shown by the orange points). Panel (b) displays $\hat{E}$ for both $I_0$ options. The orange points are the same in both panels.





In addition to nitrogen species emissions from $H_2$ engines, oil vapor emissions due to engine lubrication systems are possible (Ungeheuer et al., 2022; Bier et al., 2024). This can lead to the nucleation into ultrafine oil droplets, as observed in measurements (Ungeheuer et al., 2021). Experiments by Ponsonby et al. (2024) explored the potential of lubrication oil droplets to

act as contrail ice-forming particles, revealing that these droplets are less effective as condensation nuclei than soot particles within the temperature range of 225–245 K. Nevertheless, the study showed that lubrication oil droplets can competitively deplete plume supersaturation under soot-poor conditions, as their critical supersaturation is readily achievable. In $H_2$ plumes, these ultrafine oil droplets might contribute to ice crystal formation alongside background aerosols. Further research on the properties of oil droplets, such as their size distribution and emission indices, is necessary to fully understand their potential

role in contrail ice crystal formation processes. Yu et al. (2024) highlight that UFPs might significantly contribute to the total number of contrail ice particles if the sizes of the soot particles are reduced. Hence, understanding the role of UFPs in the contrail formation process requires the consideration of both soot emissions and particle sizes. The UFP number could potentially exceed that of emitted soot particles per flight distance, leading to a larger number of nucleated ice crystals in $H_2$ plumes compared to conventional kerosene plumes (Kärcher et al., 2015; Bier et al., 2024). This justifies our approach of increasing

the initial number of ice crystals to capture possible scenarios of additional ice-forming particle sources.

Furthermore, this study addresses a previously unexplored area by analyzing contrail evolution during the vortex phase at temperatures $> 225$ K. At higher ambient temperatures, the number of nucleated ice crystals decreases, increasing the number of ice crystals that survive the vortex phase (Bier and Burkhardt, 2022). Since our study design is independent of the specific formation processes, it hypothetically determines the survival fractions for a given fixed $N_0$.

We revised the original formulation of the vortex loss parametrization by incorporating the conservation of mass mixing ratios, adding a layer of complexity to the equations but ensuring that the physical assumptions are accurately represented. With appropriately chosen parameters, the arc tangent curve fits the data points well, both from the U2016 and present studies. Additionally, the new formulation of the fitting function is designed to account for scenarios where $N_0$ is significantly upscaled. Although the parametrization is based on the $N_0$-scaling study spanning four orders of magnitude, it remains applicable even

for more extreme values of $N_0$. The parametrization also yields plausible results in terms of high ambient temperatures. The updated parametrization does not fully capture specific data points, particularly in cases with decreased stratification and downscaled $N_0$ values. As the length scale $z_{desc}$ is the quantity that accounts for the variation of atmospheric stability, an adjustment to the formulation of $z_{desc}$, as initially proposed by Unterstrasser (2014), may be necessary to improve accuracy.

Next, we motivate our new choice for the adaptation factor $\Psi$ in the updated ice crystal loss parametrization. The original

version in U2016 was based on simulations that all had the same $EI_{H_2O}$ value, and the emission index $EI_{iceno}$ was a good estimator of the representative ice crystal size in the contrail. A straightforward generalization would be to replace $EI_{iceno}$ by the ratio $EI_{H_2O}/EI_{iceno}$ in the definition of $\Psi$. However, this quantity is only a proxy of the ice crystal mean mass ($:= I_0/N_0 = EI_{H_2O}/EI_{iceno}$) when all emitted water vapor deposits on the ice crystals, yet it neglects the contribution from the environment. Keeping in mind that the buffering effect of the two water vapor sources (from emission and from environment) is already

captured by $z_\Delta$, we now define $\Psi$ as a proxy for the typical ice crystal number concentration in the primary wake. The exact definition can be found in section A1.



One main goal of our study is to understand and systematically analyze the impact of the initial ice crystal number and of an assumed water vapor emission on the early contrail properties. We neglect any possible changes in aircraft geometry, mass, overall propulsion efficiency, etc., between current kerosene-driven aircraft and future $H_2$ aircraft. Clearly, such adaptations would affect the strength of the wake vortices and the fuel consumption (in addition to the factor $2.79^{-1}$ mentioned above). Moreover, $H_2$ fuel cells produce electric power that drives propellers. Design concepts with different numbers and positions of propellers and exhaust outlets exist in the literature. Understanding the formation and the early evolution of contrails behind a propeller-driven aircraft deserves separate modelling studies. For the time being, we prefer to limit our study to variations of $N_0$ and $I_0$ without any other aircraft adaptation in order to avoid complicating and overloading the interpretation of our results.

Once $H_2$ aircraft designs are less hypothetical, one could incorporate such adaptations (like a systematically higher aircraft mass due to larger tank systems or particular aspects of $H_2$ fuel cell aircraft) in subsequent modelling studies.

The updated parametrization is now prepared for integration into large-scale global models, where the U2016 parametrization is already implemented (Bier and Burkhardt, 2022). Bier and Burkhardt (2022) underscore the significance of the initial number of ice crystals, highlighting the critical role of our study, which focuses on the vortex phase and ice crystal loss, in assessing the climate impact of $H_2$ contrails.

## 5    Conclusions

In this study, we simulated the evolution of contrails up to an age of approximately six minutes using the EULAG-LCM model, a fluid dynamics solver coupled with a particle-based microphysical model. The focus is on contrails produced by hydrogen-powered aircraft, including both hydrogen direct combustion and hydrogen fuel cell systems. The simulations assume that ice crystal formation and vortex roll-up are complete at initialization. We varied the initial number of ice crystals over a wide range spanning four orders of magnitude, from $3.38 \times 10^{10} \, \mathrm{m}^{-1} (0.85 \times 10^{10} \, \mathrm{m}^{-1})$ to $3.38 \times 10^{14} \, \mathrm{m}^{-1} (0.85 \times 10^{14} \, \mathrm{m}^{-1})$ for an A350/B777-like (A320/B737-like) aircraft. Additionally, we scaled the emitted water vapor by a factor of 2.57 compared to the reference kerosene case. A parameter study was conducted, varying meteorological, microphysical, and aircraft parameters, focusing on the survival fraction / final number of ice crystals, a critical factor influencing subsequent contrail evolution.

Our simulations indicate that ice crystals tend to be larger in contrails from hydrogen-powered aircraft, leading to a higher likelihood of surviving the vortex phase. This effect is also observed in contrails initialized with a very small number of ice crystals, where the limited competition for available water vapor during depositional growth promotes the formation of larger ice crystals. As anticipated in previous studies, we find initial differences in ice crystal number getting reduced after the vortex phase. The higher the initial ice crystal number, the stronger this reduction is. This finding is independent of any meteorological condition. Sublimation driven by adiabatic heating in the descending vortex pair becomes more significant in drier atmospheric conditions, a trend consistent across all $N_0$ variations. Furthermore, a higher ambient temperature enhances ice crystal sublimation, though the surviving crystals tend to be larger. We find that the temperature sensitivity is particularly pronounced in the $N_0$-upscaling scenarios. A reduced thermal stratification leads to a more significant loss of ice crystals, with this effect being more pronounced at higher initial ice crystal numbers and in drier atmospheric conditions. A shift towards



a narrower initial size distribution generally produces larger ice crystal sizes, thereby reducing the extent to which initial differences are balanced, as we have consistently observed in the $H_2$ simulations. The weaker vortex descent associated with smaller aircraft results in a reduced sublimation of ice crystals in the primary wake, leading to higher survival fractions in the case of A320 aircraft. This trend is consistent across all $N_0$ values and ambient temperature scenarios.

We have refined the existing parametrization of ice crystal loss, presented by Unterstrasser (2016), by incorporating a new set

of $H_2$-contrail simulations. Given that the new simulations tend to cluster at the extremes of the parametrization function, we adjusted the fitting procedure to better capture these behaviors. The updated set of fitting coefficients is provided. A comparison with simulation results from other studies demonstrates a good agreement with our updated parametrization.

It is not yet possible to make a definitive assessment of whether flying with hydrogen is advantageous from a contrail perspective. Further research is needed to assess contrail formation processes for hydrogen propulsion, with measurement

results (Airbus, 2023) providing essential insights. As shown in Figs. 10 and 11, the initial number of ice crystals has a significant impact on the subsequent characteristics of the contrail. This study aimed to conduct a broad analysis of $N_0$ during the first six minutes of contrail evolution. Determining more robustly the number of ice crystals formed in the $H_2$ case remains an active area of research.

The analysis of the complete set of contrail-cirrus simulations (extending the study that is demonstrated in Sec. 3.4), which

are based on the $H_2$-contrail vortex phase simulations presented here, is in progress.

*Data availability.* The presented data are available from the corresponding author upon request (annemarie.lottermoser@dlr.de)

## Appendix A: Details about the ice crystal loss parametrization

### A1 Definition of the ice crystal concentration $n_0$ in the new parametrization

As described in Sec. 3.3, the adaptation factor $\Psi$ accounting for the mean ice crystal size (in the original formulation given by

$1/\mathrm{EI}^*_{\mathrm{iceno}}$) is now defined differently. In the new formulation, $\Psi$ is expressed as $\Psi = 1/n_0^*$, where $n_0$ represents an (intermediate) ice crystal number concentration:

$$n_0 = N_0/A_{\mathrm{p}}, \tag{A1}$$

where $N_0$ is the initial ice crystal number (per meter of flight path) and $A_{\mathrm{p}}$ is the intermediate plume cross-sectional area. Hence, $n_0$ represents an average concentration in the primary wake. We recall $N_0 = m_{\mathrm{C}} \times \mathrm{EI}_{\mathrm{iceno}}$ (see Eq. (1)) and the empirical

relationships (for conventional aircraft designs of different sizes as studied in Unterstrasser and Görsch (2014) and derived in U2016): $m_{\mathrm{C}} \sim b_{\mathrm{span}}{}^2$ (Eq. (A9) in U2016) and $A_{\mathrm{p}} \sim b_{\mathrm{span}}{}^2$ (Eqs. (A6) and (A7) in U2016), where $b_{\mathrm{span}}$ is the wing span of the aircraft. Hence, $n_0$ is roughly independent of aircraft type, but depends linearly on $\mathrm{EI}_{\mathrm{iceno}}$. Note that the empirical relation for $A_{\mathrm{p}}$ is derived from the simulation results and is not supposed to be adapted. The actual fuel consumption $m_{\mathrm{C}}$ may, however, deviate from the empirical relationship (representing typical kerosene fuels) and should serve as input to the parametrization.





The parametrization was trained such that the kerosene A350 reference simulations serve as reference case for determining $n_{0,\mathrm{ref}}$ and $n_0^* = n_0/n_{0,\mathrm{ref}}$ (with $N_{0,\mathrm{ref}} = 3.38 \times 10^{12}\,\mathrm{m}^{-1}$ and $b_{\mathrm{span,ref}} = 60.3\,\mathrm{m}$). Note that in particular the $m_{\mathrm{C}}$ values of our A350 setups deviate from the empirical relationship (see Fig. A1 (c) in U2016), and hence, switching to a smaller A320 aircraft gives a smaller $n_0$ value ($1.29 \times 10^9\,\mathrm{m}^{-3}$ (A350) vs. $8.93 \times 10^8\,\mathrm{m}^{-3}$ (A320)) leading to $n_0^* = 0.69$.

### A2 Differences between the original and new parametrization implementation

This section summarizes the differences between the original parametrization provided in Unterstrasser (2016) and the new one presented in Sec. 3.3. The overall design of the parametrization has not changed, and the cookbook of individual computations, as listed in Sec. A6 of U2016, received only minor changes, which are as follows:

- In step 2, the formula for computing the plume area is now $A_{\mathrm{p}} = 2 \times \pi r_{\mathrm{p}}{}^2$, which replaces Eq. (A7) of U2016.

- In step 3, $z_{\mathrm{atm}}$ and $z_{\mathrm{emit}}$ are redefined using now the adiabatic index $\kappa$ (as given in Eqs. (6) and (7)). Moreover, the
bisection method for solving the non-linear equation can be replaced by analytically defined fit functions provided in the subsequent subsection.

- In steps 3 and 4, new values for the fitting coefficients are used as given in Eqs. (13a)-(13g). Moreover, the adaptation factor $\Psi$ uses a new definition as given in subsection A1.

The supplement contains Fortran and Python implementations of the new and the original parametrization.

### A3 Analytical fit functions of $z_{\mathrm{atm}}$ and $z_{\mathrm{emit}}$

To compute the length scales $z_{\mathrm{atm}}$ and $z_{\mathrm{emit}}$ (Eqs. (6) and (7)), we employed so far the numerical bisection method. Here, we present an alternative approach that directly calculates these length scales based on input data for temperature, ambient supersaturation, and water vapor concentration. The corresponding formulae are as follows:

$$\tilde{z}_{\mathrm{atm}} = 607.46\,\mathrm{m} \times s_{\mathrm{i}}{}^{0.897} \times \left( \frac{T_{\mathrm{CA}}}{205\,\mathrm{K}} \right)^{2.225} \tag{A2}$$

and

$$\begin{aligned}
\tilde{z}_{\mathrm{emit}} = {} & 1106.6\,\mathrm{m} \times \left( \frac{\rho_{\mathrm{emit}}}{10\,\mathrm{mg/m^3}} \right)^{0.678 + 0.0116\,T_{205}} \\
& \times \exp\left( (-(0.0807 + 0.000428\,T_{205})\,T_{205}) \right)
\end{aligned} \tag{A3}$$

with $T_{205} = T_{\mathrm{CA}}/\mathrm{K} - 205$.

The resulting length scales show only slight deviations from those derived using the bisection method, with a maximum deviation of $3\,\mathrm{m}$ for $z_{\mathrm{atm}}$ and $7\,\mathrm{m}$ for $z_{\mathrm{emit}}$. Applying the analytical relations to calculate the parametrized survival fraction, we
observe no change in $44\,\%$ of the data (when rounded to two digits, as done in Tab. A1) and a maximum deviation of $2.0\,\%$.



A detailed comparison is provided in Tab. S1. The supplement contains further plots that demonstrate the suitability of the fit functions.

*Supplement.* The supplement related to this article is available online at:

.

*Author contributions.* AL performed the simulations, created the tables and figures, and wrote the manuscript with contributions from SU. AL and SU conceptualized the study, evaluated and interpreted the results, and worked on the parametrization extension.

*Competing interests.* The authors declare that they have no conflict of interest.

*Acknowledgements.* This work contributes to the DLR-internal projects H2CONTRAIL and H2EAT. Moreover, both authors received funding from Airbus SAS in the framework of understanding contrails from hydrogen propulsion. The authors thank Klaus Gierens for an
internal review of the paper draft. This work used resources of the Deutsches Klimarechenzentrum (DKRZ) granted by its Scientific Steering Committee (WLA) under project ID bd0832.





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



| Nr | AC | $T_\text{amb}$ (K) | $RH_\text{i,amb}$ (%) | $N_\text{BV}$ ($10^{-2}$ s$^{-1}$) | $N_0$ #sim | $I_0$ (g m$^{-1}$) | $r_\text{SD}$ | $f_{N,s}$ | $f_{N,s}$ | $z_\text{atm}$ (m) | $z_\text{emit}$ (m) | $\hat{z}_\text{desc}$ (m) |
|---|---|---|---|---|---|---|---|---|---|---|---|---|
| 1,2,3,4,5 | A350 | 217 | 120 | 1.15 | 5 | 15.0 | 3.0 | 0.05, 0.23, 0.65, 0.89, 0.98 | 0.06, 0.23, 0.6, 0.87, 0.97 | 164 | 249 | 339 |
| 6,7,8,9,10 | A350 | 217 | 110 | 1.15 | 5 | 15.0 | 3.0 | 0.02, 0.1, 0.28, 0.46, 0.71 | 0.0, 0.06, 0.22, 0.58, 0.86 | 85 | 249 | 339 |
| 11,12,13,14,15,16 | A350 | 217 | 120 | 1.15 | 6 | 38.55 | 3.0 | 0.02, 0.09, 0.39, 0.85, 0.98, 1.0 | 0.05, 0.2, 0.57, 0.85, 0.96, 1.0 | 164 | 546 | 339 |
| 17,18,19,20,21 | A350 | 217 | 110 | 1.15 | 5 | 38.55 | 3.0 | 0.07, 0.27, 0.63, 0.81, 0.92 | 0.08, 0.29, 0.68, 0.9, 0.98 | 85 | 546 | 339 |
| 22,23,24 | A350 | 217 | 120 | 1.15 | 3 | 15.0 | 1.0 | 0.07, 0.88, 1.0 | 0.06, 0.6, 0.97 | 164 | 249 | 339 |
| 25,26,27 | A350 | 217 | 120 | 1.15 | 3 | 15.0 | 4.0 | 0.05, 0.61, 0.98 | 0.06, 0.6, 0.97 | 164 | 249 | 339 |
| 28,29,30 | A350 | 217 | 110 | 1.15 | 3 | 15.0 | 1.0 | 0.05, 0.49, 0.93 | 0.0, 0.21, 0.86 | 85 | 249 | 339 |
| 31,32,33 | A350 | 217 | 110 | 1.15 | 3 | 15.0 | 4.0 | 0.02, 0.26, 0.67 | 0.0, 0.22, 0.86 | 85 | 249 | 339 |
| 34,35,36 | A350 | 217 | 120 | 1.15 | 3 | 38.55 | 1.0 | 0.18, 1.0, 1.0 | 0.2, 0.85, 1.0 | 164 | 546 | 339 |
| 37,38,39 | A350 | 217 | 120 | 1.15 | 3 | 38.55 | 4.0 | 0.09, 0.81, 0.99 | 0.2, 0.85, 1.0 | 164 | 546 | 339 |
| 40,41,42 | A350 | 217 | 110 | 1.15 | 3 | 38.55 | 1.0 | 0.18, 0.92, 1.0 | 0.08, 0.68, 0.98 | 85 | 546 | 339 |
| 43,44,45 | A350 | 217 | 110 | 1.15 | 3 | 38.55 | 4.0 | 0.06, 0.58, 0.89 | 0.08, 0.68, 0.98 | 85 | 546 | 339 |
| 46,47,48 | A350 | 217 | 120 | 0.5 | 3 | 15.0 | 3.0 | 0.02, 0.28, 0.66 | 0.0, 0.26, 0.94 | 164 | 249 | 515 |
| 49,50,51 | A350 | 217 | 110 | 0.5 | 3 | 15.0 | 3.0 | 0.01, 0.1, 0.31 | 0.0, 0.02, 0.73 | 85 | 249 | 515 |
| 52,53,54 | A350 | 217 | 120 | 0.5 | 3 | 38.55 | 3.0 | 0.04, 0.46, 0.84 | 0.02, 0.71, 1.0 | 164 | 546 | 515 |
| 55,56,57 | A350 | 217 | 110 | 0.5 | 3 | 38.55 | 3.0 | 0.02, 0.26, 0.53 | 0.0, 0.36, 0.96 | 85 | 546 | 515 |
| *Simulations at higher ambient temperatures* | | | | | | | | | | | | |
| 58,59,60,61,62 | A350 | 225 | 120 | 1.15 | 5 | 15.0 | 3.0 | 0.02, 0.12, 0.45, 0.76, 0.95 | 0.03, 0.14, 0.44, 0.79, 0.94 | 177 | 110 | 339 |
| 63,64,65,66,67 | A350 | 225 | 110 | 1.15 | 5 | 15.0 | 3.0 | 0.01, 0.04, 0.13, 0.25, 0.45 | 0.0, 0.01, 0.09, 0.3, 0.69 | 92 | 110 | 339 |
| 68,69,70,71,72 | A350 | 225 | 120 | 1.15 | 5 | 38.55 | 3.0 | 0.04, 0.21, 0.63, 0.9, 0.99 | 0.08, 0.28, 0.67, 0.9, 0.98 | 177 | 262 | 339 |
| 73,74,75,76,77 | A350 | 225 | 110 | 1.15 | 5 | 38.55 | 3.0 | 0.02, 0.1, 0.29, 0.46, 0.68 | 0.0, 0.07, 0.27, 0.65, 0.89 | 92 | 262 | 339 |
| 78,79,80,81,82 | A350 | 230 | 120 | 1.15 | 5 | 38.55 | 3.0 | 0.03, 0.15, 0.51, 0.83, 0.97 | 0.05, 0.21, 0.57, 0.86, 0.97 | 186 | 163 | 339 |
| 83,84,85,86,87 | A350 | 230 | 110 | 1.15 | 5 | 38.55 | 3.0 | 0.01, 0.05, 0.17, 0.31, 0.52 | 0.0, 0.03, 0.15, 0.46, 0.81 | 97 | 163 | 339 |
| 88,89,90,91,92 | A350 | 233 | 120 | 1.15 | 5 | 38.55 | 3.0 | 0.02, 0.12, 0.45, 0.78, 0.97 | 0.05, 0.19, 0.53, 0.84, 0.96 | 191 | 123 | 339 |
| 93,94,95,96,97 | A350 | 233 | 110 | 1.15 | 5 | 38.55 | 3.0 | 0.01, 0.04, 0.14, 0.26, 0.45 | 0.0, 0.02, 0.12, 0.38, 0.76 | 191 | 123 | 339 |
| 98,99,100 | A350 | 233 | 120 | 1.15 | 3 | 38.55 | 1.0 | 0.03, 0.63, 1.0 | 0.04, 0.53, 0.96 | 191 | 123 | 339 |
| 101,102,103 | A350 | 233 | 120 | 1.15 | 3 | 38.55 | 4.0 | 0.02, 0.43, 0.96 | 0.05, 0.53, 0.96 | 191 | 123 | 339 |
| 104,105,106 | A350 | 233 | 110 | 1.15 | 3 | 38.55 | 1.0 | 0.01, 0.25, 0.59 | 0.0, 0.12, 0.76 | 99 | 123 | 339 |
| 107,108,109 | A350 | 233 | 110 | 1.15 | 3 | 38.55 | 4.0 | 0.01, 0.12, 0.43 | 0.0, 0.12, 0.76 | 99 | 123 | 339 |
| 110,111,112,113,114 | A350 | 235 | 120 | 1.15 | 5 | 38.55 | 3.0 | 0.02, 0.11, 0.43, 0.75, 0.95 | 0.04, 0.18, 0.52, 0.83, 0.96 | 195 | 102 | 339 |
| 115,116,117,118,119 | A350 | 235 | 110 | 1.15 | 5 | 38.55 | 3.0 | 0.01, 0.03, 0.12, 0.23, 0.42 | 0.0, 0.01, 0.1, 0.34, 0.73 | 101 | 102 | 339 |
| *Simulations with A320/B737-like aircraft* | | | | | | | | | | | | |
| 120 | A320 | 217 | 120 | 1.15 | 1 | 3.7 | 3.0 | 0.89 | 0.72 | 164 | 176 | 231 |
| 121,122,123 | A320 | 225 | 120 | 1.15 | 3 | 3.7 | 3.0 | 0.05, 0.78, 1.0 | 0.13, 0.64, 0.96 | 177 | 76 | 231 |
| 124,125,126 | A320 | 225 | 110 | 1.15 | 3 | 3.7 | 3.0 | 0.01, 0.29, 0.86 | 0.03, 0.21, 0.78 | 92 | 76 | 231 |
| 127,128,129 | A320 | 225 | 120 | 1.15 | 3 | 9.51 | 3.0 | 0.07, 0.85, 1.0 | 0.2, 0.76, 0.99 | 177 | 185 | 231 |
| 130,131,132 | A320 | 225 | 110 | 1.15 | 3 | 9.51 | 3.0 | 0.03, 0.52, 0.96 | 0.07, 0.39, 0.9 | 92 | 185 | 231 |
| 133,134,135 | A320 | 230 | 120 | 1.15 | 3 | 9.51 | 3.0 | 0.05, 0.8, 1.0 | 0.17, 0.72, 0.98 | 186 | 114 | 231 |
| 136,137,138 | A320 | 230 | 120 | 1.15 | 3 | 9.51 | 3.0 | 0.02, 0.35, 0.88 | 0.04, 0.29, 0.85 | 186 | 114 | 231 |
| 139,140,141 | A320 | 233 | 120 | 1.15 | 3 | 9.51 | 3.0 | 0.04, 0.78, 1.0 | 0.16, 0.71, 0.97 | 191 | 85 | 231 |
| 142,143,144 | A320 | 233 | 110 | 1.15 | 3 | 9.51 | 3.0 | 0.01, 0.3, 0.86 | 0.04, 0.26, 0.82 | 99 | 85 | 231 |
| 145,146,147 | A320 | 235 | 120 | 1.15 | 3 | 9.51 | 3.0 | 0.04, 0.76, 1.0 | 0.16, 0.7, 0.97 | 195 | 70 | 231 |
| 148,149,150 | A320 | 235 | 110 | 1.15 | 3 | 9.51 | 3.0 | 0.01, 0.27, 0.84 | 0.03, 0.24, 0.81 | 101 | 70 | 231 |

**Table A1.** Summary of the simulations performed. Columns 2–5 list the meteorological parameters, while columns 6–8 present the microphysical initialization settings. Columns 8 and 9 display both the simulated and parametrized survival fractions. Lastly, columns 10–12 specify the length scales employed in the parametrization. Rows displaying five simulations correspond to sets with $N_0$-scaling factors of 100, 10, 1, 0.1, and 0.01. Rows showing three simulations represent sets with scaling factors of 100, 1, and 0.01. In the third row, a total of six simulations are included, with simulation 11 performed using a scaling factor of 1000.