# Peer review of "High-resolution modelling of early contrail evolution from hydrogen-powered aircraft"

_EGUsphere, 2024_

## Referee Comment (RC1)

Review of High-resolution modelling of early evolution from hydrogen-powered aircraft.

General comment:

The aim of this paper is to present a new parametrization for ice crystal lost during contrail vortex phase. More than just a parametrization, a lot of 3D simulation has been performed in order to have a sensitivity analysis to the apparent ice crystal number emission index and to water emission index in order to include what could be expected from a H2 combustion engines or fuel cells. This has been made by increasing or decreasing the number of ice crystals from the one of a usual kerosene jet engine by 1 and 2 order of magnitude and using the adapt water emission for an H2 engine. I think this approach is totally valid considering the lake of knowledge on H2 contrails.

First, the computational methodology is described in detailed. Then a physical analysis of the results is made with several sensitivity analysis bringing good understanding of the physics. Then the parametrization is presented and compared to their CFD results (from this study and from a past one). The comparison between the parametrization and the CFD results are convincing and the paper shows an improvement compared to the previous parametrization.

On the improvement of the parametrization by itself, I think the paper present good results which can justify by itself the publication. The paper is well written and gives a lot of information. However, I think it would gain in clarity for the reader with less knowledge if some information on the past parametrization are given. I have regroup what, in my opinion could be added in order to improve the paper into the minor comments section of the present document.

My main concerned are about the CFD by themselves that are may be in a too small domain, see mainly the major comment 1,2 and 5. In addition the considered initial condition may also introduced a modification of the survival ice crystals number (major comment 3). The major comment 4 is mainly here to increase clarity but, in my opinion, the information is important to be given. Therefor I will recommend 2 or 3 more computations. 1 with the same initial condition but with a bigger computational domain, made in such a way to leave more space above the contrail at the end or the vortex phase (comment 5) and more space at the left and right, in order to go under 1% of error on the vortex descent velocity (comment 1). Another one with your initial mesh with a gaussian plume surrounding the vortex core (almost no ice crystals in the vortex core). And one with a thinner discretization in order to the if there is no too much numerical dissipation in the vortices could be a good addtion. The comparison of the contrail evolution and the ice crystal survival fraction could be made in order to evaluate these influences. Since, I think, the paper is long enough, this can be added in an annex with a reference to it in the core text. In addition, I will say that the considered case for these simulations should be where the survival ice crystal fraction has an evolution so around a zd between 1 to 2.

On that regard I do not recommend publication on that stage, even if I think that this paper present great quality. I just want to make sure that no surprise arrive when using a better mesh, and with another choice of initialization, and since Comment 1 and 3 could bring opposite effect on the survival ice crystal number, it could be great to see if and at what degree.

Major comments:

1. For the A350 you have a domain width of 384m. Assuming periodic boundary condition, it means that at initial time, you have an infinity (let assume 6, the 2 computed and the images at the left and at the right) vortices lines up on the x axis. Using point like vortex theory, you can compute the vortex descent velocity in the 6 vortices case and compare it to the 2

vortices that you want to compute. If $\Gamma$ is the circulation of the vortex (520 m²/s), b the separation length of your 2 vortices and L the size of your computational domain (384m), the theoretical descent velocity of the vortex pair is given by $V_t = \frac{\Gamma}{2\pi b}$ wherease, taking into account the 4 other vortices, the descent velocity in your computation should be given by $V_c = \frac{\Gamma}{2\pi b} - \frac{\Gamma}{2\pi\left(L-\frac{b}{2}\right)} + \frac{\Gamma}{2\pi\left(L+\frac{b}{2}\right)} - \frac{\Gamma}{2\pi L} + \frac{\Gamma}{2\pi(L+b)}$ . (I don't think so but I may have make some mistakes so please validate or not this formula). Using your values I find Vc=1.7m/s and Vt=1.75m/s, this mean an error of 3%. In https://doi.org/10.5194/acp-15-7369-2015, the authors use a 1km domain and not only 384m (error of 0.5%). The span of their aircraft is kind of the same, however the circulation is a bit higher, still the relative error doesn't depend of the circulation. Since the descent of the vortices may affect the position of your ice crystals, it may also affect the ice crystal loss and therefore your parametrization. Your comparison with this mentioned paper show that your parametrization is always under their results and only two cases of Lewellen 2014 is under and he also has an error in descent velocity under 1%. Could you look at the influence of the domain size in transvers direction by doing a simulation with 1km in the reference stratification case and a configuration with a zd around one or 2 (it seems to be most challenging).

2. Your mesh resolution is 4 to 5 points in the core radius of the vortices (as in https://doi.org/10.5194/acp-15-7369-2015). In https://doi:10.1017/S002211209600849X it would be more 10 points and in https://doi.org/10.1063/1.4934350 it is about 8 meaning about twice as much points by radius core as yours. Have you done (for this paper or in the past) a sensitivity analysis to this parameter?

3. The discs present a constant ice crystal number, however since the ice crystals are created outside the vortex core it will take some time for them to penetrate the vortex core since diffusion is probably the main process which can allow them to penetrate the core due to the closed streamlines. For the same reason the ice crystals initially inside the vortex core are mainly trapped and will descent with the vortices. By this initialization choice, don't you increase artificially the number of ice crystals that remains in the primary wake and then the number of particles that sublimate? (As an example https://doi.org/10.1103/PhysRevFluids.8.114702 , shows (in 2D case) that the position of the initial plume can have an influence on the vertical spreading of the contrail, and putting them inside the vortex core force them to stay inside the vortex core.)

4. You gives different quantities such as $P_{amb}, T_{amb}, RH_i, amb$ and a Brundt Vaisala frequency. I guess the given value is at flight altitude whereas the pressure field follow the hydrostatic equation, and T in the atmosphere follow a constant Brundt Vaisala frequency? If so I guess it worth to be written for the non-specialist readers. Moreover, the choice of the relative humidity evolution with altitude is not trivial, since the mass fraction of water can fluctuate in order to impose a constant relative humidity with altitude, or one can keep the mass fraction of water vapor constant, or some other choice can be made… Please indicate what has been your choice in that matter.

5. In fig 2 the ice crystals are going above the flight altitude. In the picture, it seems to go dangerously close to the boundary of the domain. Are the ice crystals stopped in their movement due to the presence of the boundary condition?

Minor comments:

1. In lines 102-103 you say that in the LCM solver you use numerical particles which represent a certain number of physical particles. However I haven't found in the text how many numerical particles are used and how many physical particles are represented by them. Since you make a sensitivity analysis on the number of particles, is it the number of numerical particles which is reduced or their weigh (number of physical particles inside a numerical one). Moreover can you provide a reference on the number of numerical particles to have a good contrail representation? (may be this one: https://doi:10.5194/gmd-7-695-2014 ?)

2. Line 105 you explain that some routine of LCM has been switched of such as aggregation and radiation. In 450 you add that sedimentation was also off for vortex phase. I think it worth noticing it also in line 105. Can you also provide either some reference or some order of magnitude in order to justify that you neglect tem?

3. In the paragraph from line 114 up to 123, you gives detailed of your mesh and boundary condition. You say that you use rigid condition for the vertical direction. At first glanced I have not really understood if you meant the xmin xmax (vertical normal) boundary or the zmin, zmax boundary (vertical plans). Please could you rewrite the lines 121 to 123 such that it will be blatant.

4. You provide a lot of information about the initialization such as the plume radius which is represented by two discs. I have not find the position of the center of this discs. I imagine that it is center on the vortex center but please add the information in the text.

5. In the z direction, you have 600m which seems to go from -500m to 100m (on figure 2). Please put this information into the text.

6. In your analysis you decrease the Brundt Vaisala to 0.005s-1. In https://doi.org/10.5194/gmd-5-543-2012 gives a typical range between 0.01 to 0.03. Is your parametrization has been tested for higher stratification levels?

7. In your part 3.3 and in the annex you talk about the paper U2016 and tell this formula replace this formula from U2016 … I think the paper will gain in clarity if you write the formula from U2016

8. In formula 6 and 7 you have $T_{CA}$, $e_s$, $s_i$ which seems undefined in the text (unless I miss them). I guess they are defined in U2016, but I think you should defined them here too.

---

## Author Comment (AC1)

**Reply to reviews of manuscript „High-resolution modelling of early contrail formation behind hydrogen-powered aircraft"**

We thank both reviewers for their thoughtful and constructive comments. Their feedback stimulated valuable internal discussions regarding the approaches used in this study. We hope that incorporating the aspects highlighted by the reviewers has helped improve the manuscript's overall quality. Below, we address all points raised by the reviewers and provide additional information where necessary. The reviewer's comments are in normal font, *and our replies are in italics*. Changes in the manuscript text are in green.

*As a first step, we include the figure and accompanying section on the evolution of temperature and humidity in the (x,z)-plane, as suggested by the editor Dr. Yu. The following figure and paragraph have been added to the supplementary material:*

[Figure]

**Figure S1.** Temporal evolution of relative humidity with respect to ice (a) and temperature difference (b) in the $x, z$-plane (averaged along flight direction). In (b), the vertical background temperature profile is subtracted to highlight the temperature evolution, particularly within the primary wake. Note that the first column corresponds to 0.1 min, whereas the zeroth time step is shown in the main body of the manuscript. This distinction is made because the thermodynamic fields exhibit almost no visible features at the initial time step. The depicted exemplary simulation is performed for an A350/B777-like aircraft at $T_{\text{CA}} = 217\,\text{K}$, $RH_{\text{i,amb}} = 120\,\%$, $N_{\text{BV}} = 1.5 \times 10^{-2}\,\text{s}^{-1}$, $N_0 = 3.38 \times 10^{12}\,\text{m}^{-1}$, $I_0 = 15.0\,\text{g}\,\text{m}^{-1}$, and $r_{\text{SD}} = 3.0$.

*Figure S1 illustrates the evolution of ice relative humidity (a) and temperature difference (b), where the background temperature profile has been subtracted. Within the wake vortices, the ice relative humidity remains close to saturation until the vortices dissipate. This phenomenon results from the sublimation of ice crystals trapped in the primary wake. The sublimation increases the local water vapor concentration, balancing the decrease in RH$_i$ caused by adiabatic heating. Note that humidity values below 100% may occur as the sublimation does not instantaneously relax the humidity field to saturation. In the secondary wake, RH$_i$ also remains near saturation as detrained ice crystals deplete ambient moisture, reducing the environmental supersaturation toward saturation. The presented humidity fields are quite smooth as they are averages along flight direction.*
*Initially, the temperature perturbation is nearly everywhere close to zero. The centers of the wake vortices feature a pressure drop to compensate for centrifugal forces and can lead to a very localized temperature drop. Due to the prescribed stable stratification, air masses at the original flight altitude have a higher potential temperature than air masses beneath. Hence, the descending primary wake is identified by positive ΔT values.*

*A similar consideration explains the negative ΔT values after six minutes around z=0 m. Air masses from the primary wake rise back to the initial altitude and push or contain also air masses from lower altitudes (with lower potential temperature) to z=0 m leading to ΔT<0 K.*

*We noticed that both reviews primarily focus on the representation of dynamics in our model. It is important to emphasize that a series of sensitivity studies—examining grid resolution, initialization, and wake vortex dynamics—have already been conducted and published in previous works (Unterstrasser et al., 2014, ACP; Unterstrasser et al. 2014, JGR; Unterstrasser & Görsch 2014, JGR, Unterstrasser & Stephan, 2020). In the current study, we adopt the same configuration with respect to domain size, resolution, and wake vortex characteristics, while focusing on a detailed investigation of microphysical variations. Importantly, the dynamical setup remains unchanged from these earlier studies.*

*We also stress that the resolution requirements depend on the application. For example, in studies of hazardous wake vortex encounters, the erosion of the vortex core is critical, as it influences peak velocities and rolling moments. In contrast, for contrail simulations, the key dynamical feature is the final vertical displacement of the vortex system, which determines the vertical extent of the contrail and the magnitude of adiabatic heating.*

*The robustness of our dynamical model has been addressed in several previous studies. For instance, comparisons between NTMIX, a higher-order code, and our model EULAG-LCM show very similar results for the vertical tracer distribution after wake vortex breakup, when thermal stability and ambient turbulence are varied (Unterstrasser et al., 2014, ACP, Figure 15). A broader intercomparison of contrail LES models (Unterstrasser, 2016, Figure 13) showed excellent agreement in vortex descent predictions, with discrepancies limited to one outlier model in terms of ice crystal loss.*

*A more recent study with EULAG-LCM focused on the early contrail evolution behind a formation of two aircraft. In this application, the flow field is characterized by complex interactions between four wing-tip vortices, and the dynamical patterns differ strongly from the classical scenario of a descending wake vortex pair behind a single aircraft. In that study (Unterstrasser & Stephan (2020)), grid sensitivity studies and intermodel comparisons were conducted to evaluate the dynamical part of the simulations. EULAG-LCM was compared to MGLET, a model widely used in wake vortex studies (e.g. Misaka et al., 2012; Misaka et al., 2015, Stephan et al., 2017), and a good agreement was found. Importantly, sensitivity studies on grid resolution have also been conducted. For example, simulations with longitudinal resolutions (along flight direction) of 2 m, 1 m, and 0.5 m (Unterstrasser & Stephan, 2020) showed only minor differences in ice crystal profiles, with survival fractions differing by less than 5%.*

*These points demonstrate our awareness of testing the robustness of the dynamical model. Sensitivity analyses on grid resolution have not only been performed, but EULAG has also been benchmarked against other established LES models to validate its performance. Nonetheless, as mesh resolution was among the primary concerns raised in the review, we repeated one of the simulations with a finer grid. Additionally, a sensitivity analysis of the domain width was carried out in response to a related reviewer comment. The results confirm that both the original resolution and domain width are sufficient to capture the relevant dynamics. The outcomes of these additional sensitivity tests are included in the supplement, and a more detailed discussion is provided below.*

*Additionally, minor revisions, primarily related to grammar and spelling, have been made to the manuscript. While these changes are not individually detailed in this document, they are fully visible in the manuscript version with tracked changes.*

**Review 1**

*Dear reviewer,*

*We are grateful for your comments and suggestions. Your feedback has been very helpful in improving the quality of our paper. We hope that the new version adequately addresses your concerns. Below, we explain how we tackled your comments.*

General comment:
The aim of this paper is to present a new parametrization for ice crystal lost during contrail vortex phase. More than just a parametrization, a lot of 3D simulation has been performed in order to have a sensitivity analysis to the apparent ice crystal number emission index and to water emission index in order to include what could be expected from a H2 combustion engines or fuel cells. This has been made by increasing or decreasing the number of ice crystals from the one of a usual kerosene jet engine by 1 and 2 order of magnitude and using the adapt water emission for an H2 engine. I think this approach is totally valid considering the lake of knowledge on H2 contrails.
First, the computational methodology is described in detailed. Then a physical analysis of the results is made with several sensitivity analysis bringing good understanding of the physics. Then the parametrization is presented and compared to their CFD results (from this study and from a past one). The comparison between the parametrization and the CFD results are convincing and the paper shows an improvement compared to the previous parametrization.
On the improvement of the parametrization by itself, I think the paper present good results which can justify by itself the publication. The paper is well written and gives a lot of information. However, I think it would gain in clarity for the reader with less knowledge if some information on the past parametrization are given. I have regroup what, in my opinion could be added in order to improve the paper into the minor comments section of the present document.

> *Thank you very much for the summary of our work. We would like to emphasize that the primary objective of this study is a systematic investigation of young "hydrogen contrails" across a wide range of atmospheric parameters. The impact of the two key input parameters, initial number of ice crystals and initial water vapor emission, on the subsequent evolution into contrail-cirrus is demonstrated through exemplary cases. The updated parameterization of crystal loss is undoubtedly an important component of the manuscript.*

My main concerned are about the CFD by themselves that are may be in a too small domain, see mainly the major comment 1,2 and 5. In addition the considered initial condition may also introduced a modification of the survival ice crystals number (major comment 3). The major comment 4 is mainly here to increase clarity but, in my opinion, the information is important to be given. Therefor I will recommend 2 or 3 more computations. 1 with the same initial condition but with a bigger computational domain, made in such a way to leave more space above the contrail at the end or the vortex phase (comment 5) and more space at the left and right, in order to go under 1% of error on the vortex descent velocity (comment 1). Another one with your initial mesh with a gaussian plume surrounding the vortex core (almost no ice crystals in the vortex core). And one with a thinner discretization in order to the if there is no too much numerical dissipation in the vortices could be a good addtion. The comparison of the contrail evolution and the ice crystal survival fraction could be made in order to evaluate these influences. Since, I think, the paper is long enough, this can be added in an annex with a reference to it in the core text. In addition, I will say that the considered case for these simulations should be where the survival ice crystal fraction has an evolution so around a zd between 1 to 2.
On that regard I do not recommend publication on that stage, even if I think that this paper present great quality. I just want to make sure that no surprise arrive when using a better mesh, and with

another choice of initialization, and since Comment 1 and 3 could bring opposite effect on the survival ice crystal number, it could be great to see if and at what degree.

*Thank you for your suggestions! We address each of the points in more detail below.*

Major comments:
1. For the A350 you have a domain width of 384m. Assuming periodic boundary condition, it means that at initial time, you have an infinity (let assume 6, the 2 computed and the images at the left and at the right) vortices lines up on the x axis. Using point like vortex theory, you can compute the vortex descent velocity in the 6 vortices case and compare it to the 2 vortices that you want to compute. If $\Gamma$ is the circulation of the vortex (520 m²/s), b the separation length of your 2 vortices and L the size of your computational domain (384m), the theoretical descent velocity of the vortex pair is given by $V_t = \Gamma/2\pi b$ wherease, taking into account the 4 other vortices, the descent velocity in your computation should be given by
$V_c = \Gamma/2\pi b - \Gamma/2\pi(L-b/2) + \Gamma/2\pi(L+b/2) - \Gamma/2\pi L + \Gamma/2\pi(L+b)$ . (I don't think so but I may have make some mistakes so please validate or not this formula). Using your values I find Vc=1.7m/s and Vt=1.75m/s, this mean an error of 3%. In https://doi.org/10.5194/acp-15-7369-2015 , the authors use a 1km domain and not only 384m (error of 0.5%). The span of their aircraft is kind of the same, however the circulation is a bit higher, still the relative error doesn't depend of the circulation. Since the descent of the vortices may affect the position of your ice crystals, it may also affect the ice crystal loss and therefore your parametrization. Your comparison with this mentioned paper show that your parametrization is always under their results and only two cases of Lewellen 2014 is under and he also has an error in descent velocity under 1%. Could you look at the influence of the domain size in transvers direction by doing a simulation with 1km in the reference stratification case and a configuration with a zd around one or 2 (it seems to be most challenging).

*Your comment is greatly appreciated. We want to clarify that, in our model, the initial velocity field of the vortex pair is not continued across the boundary in the transverse direction. Therefore, no mirror vortices—of the kind you described—are present, and no initial interactions between such mirror vortices and the primary vortex pair occur. That said, due to the use of periodic boundary conditions in the transverse direction, it is true that interactions could theoretically arise if the vortices undergo strong oscillations (e.g., due to Crow instability) during the simulation. This is precisely why the simulations with lower stratification values use a larger domain in the transverse direction, as stronger vortex oscillations are observed.*
*To make sure that such cross-boundary interactions do not severely alter the final survival fraction of ice crystals, we repeated one baseline simulation with a doubled domain size in the transverse direction as you proposed. Using a larger domain width, we observe a 0.5% variation in the ice crystal survival fraction.*
*In Section 3 of the supplement, Figure S6 shows the temporal evolution of ice crystal mass and number of the reference and the larger-domain simulation [also a simulation conducted at a higher pressure level is shown]:*

[Figure]

**Figure S6.** Evolution of total ice crystal number (a) and total ice mass (b) for the reference case and three sensitivity simulations, differentiated by color and line style. The dashed black curves represent a simulation conducted at a higher ambient pressure value. The orange curves correspond to a simulation with doubled domain width. The magenta curves show the results from a simulation with a higher mesh resolution in transverse and vertical direction.

*We included the following paragraph into Section 3.2 of the supplement:*

*The default domain size in the transverse direction is 384 grid boxes, corresponding to 384 m in our reference A350/B777 simulation with a grid spacing of dx = 1 m. In a sensitivity simulation, we increase the domain width from 384 m to 768 m; the results are shown as orange curves in Fig.S6. This modification yields a slight reduction in final ice crystal survival fraction from 64.7% to 64.2% and a relative increase of 4.6% in final ice mass.*
*A plausible explanation is that, in the narrower domain, the descending vortex pairs might interact across the transverse boundaries, damping their descent and thereby enhancing ice crystal survival. Horizontal profiles reveal that, from about two minutes onward, the wider-domain simulation exhibits slightly lower ice crystal number and mass near the outer edges of the vortices, while more ice crystals and ice mass are found in the vortex centers. This supports the hypothesis that transverse interactions across the boundaries in the narrow domain might influence vortex dynamics and particle motion.*
*However, it is equally plausible that the minor deviations in the evolution of ice crystal number and mass reflect variability introduced by background turbulence, as discussed in the previous section.*
*As the wider domain has a negligible effect on the contrail properties, most notably the ice crystal survival fraction, yet significantly increases computational expense, a transverse width of 384 grid boxes is deemed sufficient for our simulations.*

*Regarding your point on the differing survival fractions compared to the LES results of Picot et al., 2015, we believe these discrepancies can be attributed to the exclusion of the Kelvin effect in their study. Including the Kelvin effect leads to lower survival fractions as the growth of small ice crystals is hampered. The importance of the Kelvin effect on contrail evolution has been mentioned in several studies (Lewellen 2012 and Lewellen 2014). Moreover, Figure 13 of Unterstrasser 2014, JGR shows substantially higher survival fractions if the Kelvin effect was deliberately switched off.*

2. Your mesh resolution is 4 to 5 points in the core radius of the vortices (as in https://doi.org/10.5194/acp-15-7369-2015. In https://doi.org10.1017/S002211209600849X it would be more 10 points and in https://doi.org/10.1063/1.4934350 it is about 8 meaning about twice as much points by radius core as yours. Have you done (for this paper or in the past) a sensitivity analysis to this parameter?

*Thank you for bringing this to our attention. A sensitivity study to the grid resolution has been performed by Unterstrasser et al., 2014, ACP, where the resolution in the flight direction was refined from 2 m to 1 m. Their results indicate that the resolution in flight direction has no significant influence on the final contrail properties. As explained above, a grid sensitivity analysis has also been performed in Unterstrasser & Stephan, 2020.*
*To specifically address the impact of grid resolution within the vortex cores, we conducted an additional sensitivity simulation in which the default mesh resolution of 1 m in both, transverse and vertical directions was refined to 0.5 m (in the A350/B777-like aircraft case). While the outcomes indicate that mesh resolution does have some influence on contrail evolution, its impact remains minor compared to other parameter variations considered in this study.*

*In the supplement, Figure S6 shows the temporal evolution of ice crystal mass and ice crystal number of the high-resolution simulation compared to the reference simulation. We included the following paragraph into Section 3.1 of the supplement:*

*In the reference setup, we employ a horizontal and vertical grid spacing of 1 m. To assess the sensitivity of contrail evolution to mesh resolution, we conduct an additional simulation using a finer resolution of 0.5 m. As shown by the magenta curves in Fig.S6, the higher-resolution simulation results in a relative increase of 7.7% in total ice mass and a reduction in ice crystal survival fraction from 64.7% to 62.1%.*
*Vertical profiles of ice crystal number and mass (not shown) indicate that, in the high-resolution setup, fewer ice crystals are detrained from the vortex system. Instead, a larger fraction remains trapped within the primary wake, where they are more prone to sublimation due to adiabatic heating.*
*The reduced detrainment can be attributed to the way secondary vorticity, generated by baroclinic torque arising from density and pressure gradients between the wake and the ambient air (Holzäpfel et al., 2001), develops in the simulation. Although the underlying physical conditions, such as pressure and density gradients, remain unchanged, the finer grid spacing enables a more accurate resolution of these instabilities, potentially altering the dynamics of vortex destabilization. We hypothesize that the improved representation of secondary vorticity results in less disruption of the vortex cores, thereby reducing ice crystal detrainment and increasing sublimation within the primary wake.*
*Alongside this physical explanation of the observed results, we note that background turbulence could likewise influence the detrainment process and contribute to the observed behavior.*
*While the resulting discrepancies in ice crystal number and mass are moderate compared to those induced by variations in the initial ice crystal number, they underscore the relevance of mesh resolution and its potential impact on simulation outcomes. However, in the context of a sensitivity study focused on variations in initial ice crystal number, where ice crystal survival fractions span the full range from 0 to 100%, we consider deviations in ice crystal survival fraction below 3% to be minor. Given the substantial computational demands of finer mesh resolutions (roughly eight times higher CPU time) and considering the comparatively minor differences in simulation outcomes, we consider the baseline resolution appropriate for the objectives of this study.*

3. The discs present a constant ice crystal number, however since the ice crystals are created outside the vortex core it will take some time for them to penetrate the vortex core since diffusion is probably the main process which can allow them to penetrate the core due to the

closed streamlines. For the same reason the ice crystals initially inside the vortex core are mainly trapped and will descent with the vortices. By this initialization choice, don't you increase artificially the number of ice crystals that remains in the primary wake and then the number of particles that sublimate? (As an example https://doi.org/10.1103/PhysRevFluids.8.114702, shows (in 2D case) that the position of the initial plume can have an influence on the vertical spreading of the contrail, and putting them inside the vortex core force them to stay inside the vortex core.)

> *Thanks for this comment. It is correct that we initialize the setup such that the discs of the ice crystals are aligned with the centers of the vortices. A sensitivity study on the initial spatial distribution of ice crystals was conducted in Unterstrasser et al., 2014, ACP. In that study, the radius of the plumes was varied (5, 15, 20, 25 m). Additionally, a ring-shaped distribution was used instead of a disc, and a non-uniform spatial distribution of ice crystals was tested. The results showed that while there are initial differences in the profiles, these differences diminish over time. Ultimately, the degree of mixing and the qualitative structure of the primary and secondary wake are very similar (except for the case with r = 5 m, which represents an extreme scenario). Stratification and relative humidity have a much stronger influence on the profiles. For this reason, we adopt the simplified initialization in the current study.*
> *Furthermore, Section 3.4 in Unterstrasser & Görsch, 2014 also examines variations in plume radius and the use of a Gaussian initialization (with either two or four plumes initially placed outside the vortex center—this is the same study as described in Unterstrasser, 2014 JGR but focuses on different aircraft types). Figure 7 there shows that Gaussian vs uniform distributions make little difference, while the aircraft type or the initial number of ice crystals has a much larger effect. These simulations are included in the ice crystal loss parameterization.*
> *The cited Saulgeot et al., 2023 study uses a 2D approach. It was compared to 3D model results, however, this was limited to an early stage of vortex stage and misses the phase where wake vortices break up due Crow instability (a vortex meandering phenomenon that cannot be resolved in 2D) and where only diffusive processes lead to a degradation of the wake vortices. Moreover, the study does not clearly demonstrate the resolution requirements for accurately predicting ice crystal loss, which is a central focus of our work. Notably, we have previously found substantial discrepancies in ice crystal loss estimates between 2D and 3D simulations (see Figures 2 and 4 in Unterstrasser, 2014, JGR).*

4. You gives different quantities such as $P_{amb}$, $T_{amb}$, $RH_i$, $amb$ and a Brundt Vaisala frequency.

I guess the given value is at flight altitude whereas the pressure field follow the hydrostatic equation, and T in the atmosphere follow a constant Brundt Vaisala frequency? If so I guess it worth to be written for the non-specialist readers. Moreover, the choice of the relative humidity evolution with altitude is not trivial, since the mass fraction of water can fluctuate in order to impose a constant relative humidity with altitude, or one can keep the mass fraction of water vapor constant, or some other choice can be made… Please indicate what has been your choice in that matter.

> *Thank you for pointing this out. In the simulation, the ambient pressure is prescribed at the lower boundary of the simulation domain. Using the hydrostatic equation and assuming a vertically constant Brunt-Väisälä frequency, the pressure profile throughout the simulation domain is calculated. Temperature is also computed based on the hydrostatic equation. The equations defining the profiles of p, $\rho_{air}$, T, and θ are given in Eq. 2 of Clark & Farley (1984). The relative humidity is prescribed as constant throughout the entire simulation domain. As a result, the absolute humidity varies from 0.1 gm⁻³ at flight altitude to 0.15 gm⁻³ at the lower boundary of the simulation domain (assuming a temperature of 217 K at flight altitude and*

*our standard values for $p_0$ and $N_{BV}$). The corresponding mass fraction ranges between 0.0003 and 0.0004.*

*To make this clearer, we changed the paragraph*

*"Additionally, we investigate the influence of different atmospheric conditions on the evolution of contrail ice crystals. At the lower boundary of the simulation domain, the pressure is 250 hPa and the air density is roughly 0.4 kgm⁻³ changing with the prescribed ambient temperature. We use a background turbulence field with an eddy dissipation rate epsilon=10⁻⁷m²s⁻³. Previous research has highlighted the importance of ambient temperature at cruise altitude, ambient relative humidity, and atmospheric stratification (described by the Brunt-Väisälä frequency) as key input parameters (Lewellen 2014a, Unterstrasser, 2016). We vary these parameters as outlined in Tab.1."*

*into*

*"Previous research has highlighted the importance of ambient temperature at cruise altitude $T_{CA}$, ambient relative humidity with respect to ice $RH_{i,amb}$, and atmospheric stratification $N_{BV}$ (described by the Brunt-Väisälä frequency) as key input parameters (Lewellen 2014a, Unterstrasser, 2016). We vary these parameters as outlined in Tab.1.*
*At the lower boundary of the simulation domain, we prescribe an air pressure of $p_0$=250 hPa and an air density of approximately $\rho_{air,0}$=0.4 kgm⁻³. For ambient temperature, we prescribe cruise-altitude values of $T_{CA}$=217, 225, 230, 233, and 235 K. The relative humidity with respect to ice is assumed to be constant throughout the entire simulation domain and set to 110 or 120%. We use a background turbulence field with an eddy dissipation rate of $\varepsilon$=10⁻⁷m²s⁻³. The pressure and temperature vertical profiles are computed for an atmosphere with a vertically constant Brunt-Väisälä frequency according to Eq.2 in Clark and Farley, 1984.*
*At cruise altitude, the resulting air pressure ranges from 231 to 235 hPa, depending on Brunt-Väisälä frequency and temperature. In Tab.1, an average air pressure value is provided. It would seem appropriate to vary ambient pressure in combination with $T_{CA}$ to reflect a change in flight altitude. However, a variation of pressure has only little impact on the simulated contrail properties as demonstrated in Sec.3.3 in the supplement. This is because the background water vapor mass concentration is independent of ambient pressure. Hence, we do not adapt $p_0$."*

5. In fig 2 the ice crystals are going above the flight altitude. In the picture, it seems to go dangerously close to the boundary of the domain. Are the ice crystals stopped in their movement due to the presence of the boundary condition?

*Thank you for this valuable comment. We prescribe rigid boundaries in vertical direction, meaning that any flux of ice crystals is prohibited across the boundaries. The issue you describe was indeed encountered in simulations with ambient temperatures of 230, 233, and 235 K, where ice crystals reached the upper boundary of the domain. To address this, we increased the vertical extent of the simulation domain—using 800 m instead of the default 600 m for the A350-like aircraft, and 634 m instead of 488 m for the A320-like aircraft. 600 grid points in the vertical should therefore be regarded as a default value, but it is adjusted depending on parameter settings such as stratification and temperature.*

*To make this aspect clearer, we included the following into Section 2.2:*

*Additionally, for high cruise-altitude temperatures (i.e. $T_{CA}$=230, 233, and 235 K), the vertical domain size was increased from 600 to 800 grid points for the A350, and from 856 to 1112 grid points for the A320.*

Minor comments:

1. In lines 102-103 you say that in the LCM solver you use numerical particles which represent a certain number of physical particles. However I haven't found in the text how many numerical particles are used and how many physical particles are represented by them. Since you make a sensitivity analysis on the number of particles, is it the number of numerical particles which is reduced or their weigh (number of physical particles inside a numerical one). Moreover can you provide a reference on the number of numerical particles to have a good contrail representation? (may be this one: https://doi:10.5194/gmd-7-695-2014*?)*

*Thank you for this thoughtful comment. You are right, we have not mentioned the role of the simulations particles when increasing the number of real ice crystals. Hence, we included the following into Section 2.1:*

*Each simulation particle (SIP) represents a certain number of real ice crystals (referred to as the weight of the SIP) with identical properties, such as size and mass.*

*The following sentence is added in Section 2.2:*

*Varying the number of ice crystals in the model is achieved by varying the weight of the SIPs, which keeps their total number roughly the same.*

*A sensitivity study regarding the number of simulation particles has been performed in Unterstrasser, 2014, ACP. There, the SIP number was varied between 23e6 and 223e6. The following plot shows the temporal evolution of SIP number (left) and ice crystal number (right). The ice crystal number evolution is not affected by the SIP number variation. Hence, we conclude that our default number of around 23xe6 SIPs is sufficient for a good contrail representation.*

[Figure]

*Moreover, Unterstrasser & Sölch (2014) conducted extensive tests on the numerical convergence properties of our particle-based approach. They demonstrate and argue why astonishingly few SIPs per grid box are needed to reach numerical convergence (note, however, that this publication analyzed contrail-cirrus simulations and not contrails during the vortex phase).*

2. Line 105 you explain that some routine of LCM has been switched of such as aggregation and radiation. In 450 you add that sedimentation was also off for vortex phase. I think it worth noticing it also in line 105. Can you also provide either some reference or some order of magnitude in order to justify that you neglect tem?

> *Thank you for the comment.*
> *Radiative processes are neglected in our simulations, as they act on significantly longer time scales (on the order of hours) compared to the time scales relevant in the vortex phase, which are on the order of a few minutes. The effect of radiation on contrail-cirrus was treated in Section 4 of Unterstrasser & Gierens, 2010 (Part 2) and in Lewellen, 2014a,b. The heating rates lead to changes over time scales of hours. However, they are negligible compared to the adiabatic heating and buoyancy effects during the vortex phase.*
> *The probability for aggregation events to happen increases with increasing differential sedimentation velocities and increasing particle sizes (Sölch & Kärcher, 2010). Aggregation becomes significant when particle size differences exceed approximately 100 µm. Since such large differences in particle size do not occur during the vortex phase in our simulations, we deliberately switch off aggregation in our study.*
> *Sedimentation is a process that we included in the vortex phase simulations. However, in Section 3.4, we mistakenly stated that sedimentation was turned off. To correct this, we have removed the following incorrect sentence from Section 3.4:*
>
>

3. In the paragraph from line 114 up to 123, you gives detailed of your mesh and boundary condition. You say that you use rigid condition for the vertical direction. At first glanced I have not really understood if you meant the xmin xmax (vertical normal) boundary or the zmin, zmax boundary (vertical plans). Please could you rewrite the lines 121 to 123 such that it will be blatant.

> *Thank you for pointing this out. With "rigid boundaries," we mean the zmin and zmax values. It means that no normal flux across the vertical boundary is allowed.*
> *We added the following sentence into Section 2.2:*
>
> *In the vertical direction, rigid boundary conditions are used, meaning that the vertical velocity component is zero at the top and bottom boundaries of the simulation domain.*

4. You provide a lot of information about the initialization such as the plume radius which is represented by two discs. I have not find the position of the center of this discs. I imagine that it is center on the vortex center but please add the information in the text.

> *Thank you for your advice, this description is indeed missing in the text. We added the following sentence into Section 3.1:*
>
> *The centers of the plumes and of the wake vortices are collocated on each side.*

5. In the z direction, you have 600m which seems to go from -500m to 100m (on figure 2). Please put this information into the text.

*Thank you for noticing that. We added this additional information in the caption of Figure 2:*

*The z-coordinate is shifted such that z=0 m corresponds to the cruise altitude (which is also done in Figs. 4, 5, and 6).*

6. In your analysis you decrease the Brundt Vaisala to 0.005s-1. In https://doi.org/10.5194/gmd-5-543-2012 gives a typical range between 0.01 to 0.03. Is your parametrization has been tested for higher stratification levels? https://doi.org/10.5194/gmd-5-543-2012 gives a typical range between 0.01 to 0.03. Is your parametrization has been tested for higher stratification levels?

*No, the default values that have been used throughout the present as well as the past studies have been 0.5e-2 and 1.15e-2 $s^{-1}$.*
*As shown in Dürbeck & Gerz, 1996, the distribution of $N_{BV}$ values peaks around 0.01$s^{-1}$ for tropospheric flight altitudes, with values exceeding 0.025 $s^{-1}$ occurring only rarely. Clearly, stratospheric $N_{BV}$ values are typically higher and peak at 0.02 $s^{-1}$. However, ice supersaturation is very rare in the stratosphere. Hence, our $N_{BV}$ selection represents tropospheric conditions.*
*Moreover, in non-dimensional terms, $N_{BV}*$ is given by $N_{BV} \times t_0$ ($t_0$ is the vortex time scale). This quantity varies over a wider range as $t_0$ changes with a variation of aircraft type. Figure 8 in Unterstrasser & Görsch (2014) depicts the non-dimensional vertical displacement $z_{desc}* = z_{desc}/b_0$ as a function of $N_{BV}*$, where data points with different $t_0$ values lie on one universal line. Hence, one may expect that the simulations cover implicitly a larger variation of $N_{BV}$.*

7. In your part 3.3 and in the annex you talk about the paper U2016 and tell this formula replace this formula from U2016 ... I think the paper will gain in clarity if you write the formula from U2016

*Thank you for this point. In the main body, we repeat the definition of $z_{desc}$ so that all length scale definitions are given. Therefore, in Section 3.3, we add*

*We repeat the definition of $z_{desc}$, which is $z_{desc} = (8\Gamma_0/(\pi\, N_{BV}))^{1/2}$.*

*In A2, we now write:*

*In step 2, the formula for computing the plume area is now $A_p = 2 \times \pi r_p^2$, which replaces Eq. (A7) of U2016, which was given by $A_p = 4 \times \pi r_p^2$.*

8. In formula 6 and 7 you have $T_{CA}$, $e_s$, $s_i$ which seems undefined in the text (unless I miss them). I guess they are defined in U2016, but I think you should defined them here too.

*Yes, you are right, thanks for noticing. Generally, we decided to change $T_{amb}$ into $T_{CA}$ to make it clearer that temperature is prescribed at cruise altitude in our setup. Therefore, we give the definition of $T_{CA}$ already in Section 2.2:*

*Previous research has highlighted the importance of ambient temperature at cruise altitude $T_{CA}$, ambient relative humidity $RH_{i,amb}$, and atmospheric stratification $N_{BV}$ (described by the Brunt-Väisälä frequency) as key input parameters.*

*Specifying $e_s$ and $s_i$, we added the following into Section 3.3:*

*The water vapor saturation pressure is represented by $e_s$, and $s_i=RH_i-1$ denotes the supersaturation.*

**Review 2**

*Dear reviewer,*

*Thank you very much for your comments and your feedback. We hope that the new version adequately addresses the points you raised. Please find below how we tackled your comments.*

General comments

This manuscript presents a numerical study of ice crystal formation behind hydrogen-powered aircraft using a 3D large-eddy simulation (LES) framework coupled with a Lagrangian microphysical model. The focus lies on the vortex phase of contrail evolution, with a comparative analysis between hydrogen- and kerosene-fueled aircraft. The study explores the sensitivity of contrail properties to ambient humidity, temperature, initial ice crystal properties, and aircraft type. A new parameterization for the survival fraction of ice crystals under hydrogen combustion scenarios is proposed and tested against the LES results.

This is a timely and relevant contribution, particularly given the increasing interest in low-emission aviation technologies. The manuscript is well written and the results are clearly presented. I believe this study is of high quality and should be published after addressing a few key issues. In particular, the mesh resolution near the vortex core appears coarse when compared to prior studies (e.g., http://dx.doi.org/10.1063/1.4807063, https://doi.org/10.1063/1.4934350), which used at least 10 grid points across the vortex core radius. I would encourage the authors to comment on this, and ideally to provide a resolution sensitivity test (e.g., for both A350 and A320 cases), particularly regarding contrail properties such as vertical profiles of ice mass and ice crystal survival fractions.

> *We greatly appreciate your comment. As stated earlier in this document and in response to a similar point raised by Reviewer 1, we repeated one of the baseline simulations using a finer mesh resolution. The outcome supports our conclusion that the originally used resolution is sufficient for the objectives of this study, which primarily focuses on hydrogen propulsion effects by varying the initial ice crystal number and water vapor emissions. The specific results and interpretation of the grid sensitivity analysis are provided in our response to Comment 2 of Reviewer 1 and detailed in Section 3 of the supplement.*

Specific comments

1. Section 2.1 (second paragraph): The Lagrangian particle framework is introduced, but it is not stated whether drag forces acting on the particles are considered. Given that the ice crystals exceed 1 μm in radius, could the authors clarify whether drag is included in the particle dynamics?

    > *Thank you for this thoughtful comment. In the current framework, drag is not considered. The effect of particle inertia, specifically in terms of centrifugal acceleration, was investigated in Unterstrasser, 2014, JGR. That study found that inertial effects may become relevant for larger particles located at plume positions exposed to the maximum tangential velocity*

*induced by the rotating vortices. In these specific cases, particles may leave the vortex system due to inertial effects. However, we believe this phenomenon affects only the largest particles, which would, anyway, likely survive adiabatic heating. Therefore, we consider the impact on the survival fraction to be negligible. Additionally, it is important to note that ice crystals captured within the trailing vortices tend to shrink over time, making them less susceptible to centrifugal forces. Moreover, as the simulation progresses, the vortex circulation—and consequently the tangential velocity—decreases, further reducing the influence of inertial effects.*

*From a computational perspective, including inertial effects would require a significantly smaller time step—on the order of $10^{-5}$ seconds compared to the $10^{-2}$ seconds used in the current framework—leading to a substantial increase in computational cost for the computation of the particles' trajectories. Given the probably limited impact of inertia on the key metric of survival fraction and considering the additional computational expense, we believe its exclusion is justified in the context of this study.*

*We included the following sentence into Section 2.1:*

*The effect of particle inertia, specifically centrifugal acceleration in the rotating wake vortices, is neglected due to the small ice crystal sizes present in the primary wake, as shown in Unterstrasser, 2014, JGR.*

2. Line 124: You mention the simulation starts "several seconds" after emission. Can you specify the exact time corresponding to the plume age? This is important for comparison with prior studies (e.g., https://doi.org/10.5194/acp-15-7369-2015).

   *We acknowledge your interest in a precise specification of the initialization time and understand the intention to assess the comparability of initial conditions across different studies. However, we believe that providing a specific value is not particularly meaningful in this context. The focus of our analysis is analyzing contrail properties after vortex break-up, and model comparisons should focus on contrail properties at that time.*

3. Line 121: Please clarify what is meant by "rigid boundary conditions". Do you refer to the vertical boundaries being reflective, fixed-value, or another type?

   *Thank you for this comment. Rigid boundary conditions mean that any flux across the vertical boundaries in normal direction is prohibited. Along the boundary surface, free slip is assumed. As pointed out in our reply to Minor comment 3 of Reviewer 1, we included an additional description into Section 2.2.*

4. Line 170: About the initial crystal distribution, I understand the lack of direct measurements of crystal properties (e.g., radius and density), but more explanation on how the initial mean radius and density were selected would be appreciated. Was this based on previous simulations (e.g., https://doi.org/10.5194/acp-24-2319-2024), theoretical estimates, or adjusted to match a specific ice number emission index?

   *Thank you for your hint that we need to specify this. Our reply is a bit verbose, as we are not sure whether density refers to the density of ice or the mass/number concentrations of the ice crystal population.*

   *In our setup, the initial total contrail ice mass $I_0$ per unit length is given by the amount of emitted water vapor, and the analogous quantity for the number is denoted as $N_0$. $I_0$ and $N_0$*

*are the main parameters that are varied in our study (besides temperature and aircraft type). From those two quantities, the mean ice crystal mass follows simply as $I_0/N_0$. The (geometric) mean radius can be computed with further assumptions on the ice crystal habit, density of ice, and shape of the size distribution. The initial mass and number concentrations simply follow from $I_0/A_0$ and $N_0/A_0$, where $A_0$ is the cross-sectional area of the initial plumes. Clearly, uniform plume concentrations are an idealization. Hence, the values of "radius and density" (we believe you refer to ice crystal mean mass and ice crystal mass/number concentration) are direct consequences of our setup parameters and choices.*

*We added the following in Section 2.2:*

*For this estimate, we use a density of ice $\rho_i = 0.92$ gcm$^{-3}$, and assume spherical particles, which is a reasonable approximation for small ice crystals with aspect ratios close to one. The radius corresponding to the mean particle mass, $r_0$, is then approximated by $\sim (I_0/(\rho_i N_0))^{1/3}$, assuming that all emitted water vapor is deposited onto the ice crystals. While the LCM represents ice crystal habits as hexagonal columns, this simplified estimate for $r_0$ serves as a first-order approximation to illustrate the general relationship between $N_0$, $I_0$, and the resulting crystal size.*

Technical corrections

1. In lines 309, should be "A320 aircraft" not "A320aircraft".
2. In lines 309, should be "with a wing span" not "with wing span".

*Thank you very much for the careful reading of the manuscript. We corrected the oversights.*

[revised manuscript text omitted]

To compute the length scales $z_\text{atm}$ and $z_\text{emit}$ (Eqs. (7) and (8)), we employed so far the numerical bisection method. Here, we present an alternative approach that directly calculates these length scales based on input data for temperature, ambient supersaturation, and water vapor concentration. The corresponding formulae are as follows:

$$\tilde{z}_\text{atm} = 607.46\,\text{m} \times s_\text{i}^{0.897} \times \left(\frac{T_\text{CA}}{205\,\text{K}}\right)^{2.225} \tag{A2}$$

and

$$\tilde{z}_\text{emit} = 1106.6\,\text{m} \times \left(\frac{\rho_\text{emit}}{10\,\text{
[revised manuscript text omitted]

**1     Temporal and spatial evolution of  temperature and   humidity during the vortex phase**

Figure S1 illustrates the evolution of ice relative humidity (a) and temperature difference (b), where the background temperature profile has been subtracted. Within the wake vortices, the ice relative humidity remains close to saturation until the vortices

5    dissipate. This phenomenon results from the sublimation of ice crystals trapped in the primary wake. The sublimation increases the local water vapor concentration, balancing the decrease in $RH_{\mathrm{i}}$ caused by adiabatic heating. Note that humidity values below $100\,\%$ may occur as the sublimation does not instantaneously relax the humidity field to saturation. In the secondary wake, $RH_{\mathrm{i}}$ also remains near saturation as detrained ice crystals deplete ambient moisture, reducing the environmental supersaturation toward saturation. The presented humidity fields are quite smooth as they are averages along flight direction.

10    Initially, the temperature perturbation is nearly everywhere close to zero. The centers of the wake vortices feature a pressure drop to compensate for centrifugal forces and can lead to a very localized temperature drop. Due to the prescribed stable stratification, air masses at the original flight altitude have a higher potential temperature than air masses beneath. Hence, the descending primary wake is identified by positive $\Delta T$ values. A similar consideration explains the negative $\Delta T$ values after six minutes around $z = 0\,\mathrm{m}$. Air masses from the primary wake rise back to the initial altitude and push or contain also air

15    masses from lower altitudes (with lower potential temperature) to $z = 0\,\mathrm{m}$ leading to $\Delta T < 0\,\mathrm{K}$.

**2    Further information on ice crystal loss parametrization**

**2.1    Evaluation of $z_{\mathrm{atm}}$ and $z_{\mathrm{emit}}$**

In the ice crystal loss parametrization, the two length scales $z_{\mathrm{atm}}$ and $z_{\mathrm{emit}}$ are implicitly defined. In the original version in U2016 (= Unterstrasser, 2016), the non-linear equations were solved using a numerical method (classical bisection method).

20    To speed up evaluations and to provide explicit formulations, fit formulae for the two length scales $z_{\mathrm{atm}}$ and $z_{\mathrm{emit}}$ were derived as outlined in Sec. (A3) of this study. In order to compare both versions simulation-wise, the length scale values determined with the bisection method and the fit formulae are compared. Moreover, the corresponding survival fractions based on either evaluation method are calculated. The outcomes are provided in Tab. S1. As  noted in the main text

[Figure]

**Figure S1.** Temporal evolution of relative humidity with respect to ice (a) and temperature difference (b) in the $x,z$-plane (averaged along flight direction). In (b), the vertical background temperature profile is subtracted to highlight the temperature evolution, particularly within the primary wake. Note that the first column corresponds to 0.1 min, whereas the zeroth time step is shown in the main body of the manuscript. This distinction is made because the thermodynamic fields exhibit almost no visible features at the initial time step. The depicted exemplary simulation is performed for an A350/B777-like aircraft at $T_{CA} = 217\,\text{K}$, $RH_{\text{i,amb}} = 120\,\%$, $N_{BV} = 1.5 \times 10^{-2}\,\text{s}^{-1}$, $N_0 = 3.38 \times 10^{12}\,\text{m}^{-1}$, $I_0 = 15.0\,\text{g\,m}^{-1}$, and $r_{SD} = 3.0$.

present study: "Applying the , applying the analytical relations to  compute the parametrized survival fraction
25   yields no change for 44 % of the data points ( rounded to two  decimal places, as in Tab. A1) , and the maximum deviation observed is 2.0 %.

**3**

**2.1 Comparison of the original and new ice crystal loss parametrization**

30  The original ice crystal loss parametrization proposed in U2016 has been implemented in several larger-scale contrail models to refine the contrail initialization in those models (Gruber et al., 2018; Bier and Burkhardt, 2022), and applications were confined to conventional kerosene contrails.

    This section presents comparison plots between the original and updated version of the ice crystal loss parametrization. Figure S2 shows scatter plots of $z_\Delta$ (panel (a)) and the parametrized survival fraction $\hat{f}_{N,s}$ (panel (b)), with the $x$-axis repre-
35  senting the original (2016) values and the $y$-axis showing the updated (2025) data. The values of $z_\Delta$ are similar across both formulations, although $z_{\Delta,2025}$ is generally slightly lower. However, differences in $z_\Delta$ should not be over-interpreted as this

[Figure]

**Figure S2.** Scatter plot comparing the 2016 $z_\Delta$ values with the $z_\Delta$ values from the present study (a), and a similar comparison for the parametrized survival fractions (b).

quantity serves as argument in an arctan-type function (see Eq. (12)) to retrieve the survival fraction. The arctan-type function formulation includes three fit coefficients that change from one to the other version. Hence, panel (b) shows the eventual differences in the parametrized survival fraction from the two versions. Likewise, $\hat{f}_{N,\mathrm{s}}$ exhibits only minor scatter between the two versions.

Furthermore, we reproduce plots that were shown in U2016 (Figs. 5, 9, and 10 in that publication). In the new versions of those plots (Figs. S3-S5 in this document), we juxtapose the outcome of the original and the new parametrization. This should demonstrate that the switch to the new formulation has only marginal implications on applications focusing on conventional kerosene contrails.

**3   Sensitivity analyses: Numerical and physical aspects**

In the following, three sensitivity studies are presented. We compare the results of a reference simulation, performed for an A350/B777-like aircraft at $T_{\mathrm{CA}} = 217\,\mathrm{K}$, $RH_{\mathrm{i,amb}} = 120\,\%$, $N_{\mathrm{BV}} = 1.5 \times 10^{-2}\,\mathrm{s}^{-1}$, $N_0 = 3.38 \times 10^{12}\,\mathrm{m}^{-1}$, $I_0 = 15.0\,\mathrm{g\,m}^{-1}$, and $r_{\mathrm{SD}} = 3.0$ (solid black lines in Fig. S6). These are compared to simulations in which the grid resolution, domain size, and ambient pressure are varied individually.

**3.1   Impact of grid resolution**

In the reference setup, we employ a horizontal and vertical grid spacing of $1\,\mathrm{m}$. To assess the sensitivity of contrail evolution to mesh resolution, we conduct an additional simulation using a finer resolution of $0.5\,\mathrm{m}$. As shown by the magenta curves in Fig. S6, the higher-resolution simulation results in a relative increase of $7.7\,\%$ in total ice mass and a reduction in ice

[Figure]

**Figure S3.** Reproduced version of Fig. 5 in U2016. The first two columns show the original plot from U2016. The third and fourth  columns use the parametrized survival fractions as obtained from the new parametrization version described in the present study.

Adapted figure caption of U2016:

*Columns 1 and 3: Relationship between simulated survival fraction $f_{N,s}$ and $z_\Delta$. The grey curve shows the fit function  $\hat{a}(z_\Delta)$ as defined in Eq. (12) in the present study.*

*Columns 2 and 4: Relationship between simulated survival fraction $f_{N,s}$ and approximated survival fraction $\hat{f}_{N,s}$. The black line shows the one-to-one line. Each row shows a subset of simulations taken from various simulation blocks defined in Table A2 of U2016. For example, the first row shows simulations of block 1, where $RH_i$ and $T_{CA}$ are varied. The legend in the plot provides a list of the symbols and colors, which uniquely  defines the  simulation parameters of each plotted data point. The root mean square of the absolute error $\hat{f}_{N,s} - f_{N,s}$ is denoted as $E_{abs}$ and given for each subset.*

[Figure]

**Figure S4.** Reproduced version of Fig. 9 in U2016. The first two rows show the original plot from U2016. The other rows use the parametrized survival fractions as obtained from the new parametrization version described in the present study evaluating $z_{atm}$ and $z_{emit}$ either via bisection (rows 3 &and 4) or by employing the fit functions (rows 5 &and 6).

Adapted figure caption of U2016:

*Sensitivity of ice crystal loss to $EI_{iceno}$ for various values of $RH_i$, T, $N_{BV}$, and b (from left to right). See legend for the color coding. Rows 1, 3, and 5: Ice crystal number per meter of flight path before and after the vortex phase (dashed and solid curves). Note that the initial ice crystal number depends only on b and $EI_{iceno}$ (following Eq. (A10) in U2016, which assumes a water vapor emission index of 1.25 –kg/kg). Hence, only one dashed curve is shown in the columns for $RH_i$, T, and $N_{BV}$, respectively. Rows 2, 4, and 6: Survival fraction.*

[Figure]

**Figure S5.** Reproduced version of Fig. 10 in U2016. The first column  shows the original plot from U2016. The two other columns use the new parametrization version (for both types of $z_{\text{atm}}$ and $z_{\text{emit}}$).

Adapted figure caption of U2016:

*Ice crystal number per meter of flight path (top) and contrail depth (bottom) after the vortex phase as a function of $RH_i$, $T$, $N_{BV}$, or $b$. $EI_{iceno}$ is  $10^{15}$ or  $10^{14}$ $kg^{-1}$. The contrail depth parametrization does not depend on $EI_{iceno}$. Note that the parametrization of the contrail depth $H$ was not updated in the present study. The slightly different results come from the fact that the parametrization of $H$ uses the parametrized $f_{N,s}$ value as input. Note that the original plot in U2016 showed an additional panel with ice crystal number concentrations, which is left out here.*

[Figure]

**Figure S6.** Evolution of total ice crystal number (a) and total ice mass (b) for the reference case and three sensitivity simulations, differentiated by color and line style. The dashed black curves represent a simulation conducted at a higher ambient pressure value. The orange curves correspond to a simulation with doubled domain width. The magenta curves show the results from a simulation with a higher mesh resolution in transverse and vertical direction.

crystal survival fraction from 64.7 % to 62.1 %. Vertical profiles of ice crystal number and mass (not shown) indicate that, in the high-resolution setup, fewer ice crystals are detrained from the vortex system. Instead, a larger fraction remains trapped within the primary wake, where they are more prone to sublimation due to adiabatic heating. The reduced detrainment can be attributed to the way secondary vorticity, generated by baroclinic torque arising from density and pressure gradients between the wake and the ambient air (Holzäpfel et al., 2001), develops in the simulation. Although the underlying physical conditions, such as pressure and density gradients, remain unchanged, the finer grid spacing enables a more accurate resolution of these instabilities, potentially altering the dynamics of vortex destabilization. We hypothesize that the improved representation of secondary vorticity results in less disruption of the vortex cores, thereby reducing ice crystal detrainment and increasing sublimation within the primary wake. Alongside this physical explanation of the observed results, we note that background turbulence could likewise influence the detrainment process and contribute to the observed behavior. While the resulting discrepancies in ice crystal number and mass are moderate compared to those induced by variations in the initial ice crystal number, they underscore the relevance of mesh resolution and its potential impact on simulation outcomes. However, in the context of a sensitivity study focused on variations in initial ice crystal number, where ice crystal survival fractions span the full range from 0 to 100 %, we consider deviations in ice crystal survival fraction below 3 % to be minor. Given the substantial computational demands of finer mesh resolutions (roughly eight times higher CPU time) and considering the comparatively minor differences in simulation outcomes, we consider the baseline resolution appropriate for the objectives of this study.

**3.2 Impact of domain size**

The default domain size in the transverse direction is 384 grid boxes, corresponding to 384 m in our reference A350/B777 simulation with a grid spacing of $dx = 1$ m. In a sensitivity simulation, we increase the domain width from 384 m to 768 m; the results are shown as orange curves in Fig. S6. This modification yields a slight reduction in final ice crystal survival fraction from 64.7 % to 64.2 % and a relative increase of 4.6 % in final ice mass. A plausible explanation is that, in the narrower domain, the descending vortex pairs might interact across the transverse boundaries, damping their descent and thereby enhancing ice crystal survival. Horizontal profiles reveal that, from about two minutes onward, the wider-domain simulation exhibits slightly lower ice crystal number and mass near the outer edges of the vortices, while more ice crystals and ice mass are found in the vortex centers. This supports the hypothesis that transverse interactions across the boundaries in the narrow domain might influence vortex dynamics and particle motion. However, it is equally plausible that the minor deviations in the evolution of ice crystal number and mass reflect variability introduced by background turbulence, as discussed in the previous section. As the wider domain has a negligible effect on the contrail properties, most notably the ice crystal survival fraction, yet significantly increases computational expense, a transverse width of 384 grid boxes is deemed sufficient for our simulations.

**3.3 Impact of pressure variation**

A variation of the ambient pressure value has only minor impact on the evolution of ice crystal mass and number, see black dashed curves in Fig. S6. The pressure at flight altitude of the reference simulation is 231 hPa. We increase the pressure to 350 hPa, keeping all other setup parameters (specifically ambient temperature) unchanged. We observe a slightly reduced

final ice mass and a slightly increased number of surviving ice crystals in the higher-pressure case. These differences can be primarily attributed to the pressure dependence of water vapor diffusivity, which appears in the governing equation for ice mass growth (Sölch and Kärcher, 2010). Since diffusivity is inversely proportional to pressure, higher ambient pressure leads to lower diffusivity, thereby reducing the rate of ice crystal growth. Conversely, sublimation is also reduced under higher pressure for the same reason, leading to a slightly higher survival fraction. An additional, though secondary, factor is the pressure dependence of sedimentation velocity. Increased pressure results in a small reduction in sedimentation velocity. However, for a pressure increase of $120\,\mathrm{hPa}$, the resulting change in sedimentation velocity is on the order of $0.5\,\%$, and is thus considered negligible in this context. Overall, the sensitivity of contrail evolution to ambient pressure variations is weak. The final ice crystal survival fraction changes from $64.7\,\%$ to $68.7\,\%$, and the total ice mass differs relatively by $4.0\,\%$. This limited sensitivity is expected: the amount of available atmospheric water vapor, expressed in terms of water vapor concentration $\rho_{\mathrm{WV,avail}} = (RH_{\mathrm{i,amb}} - 1) \times \rho_{\mathrm{WV,sat,ice}} = (RH_{\mathrm{i,amb}} - 1) \times \frac{e_{\mathrm{s}}(T)}{R_{\mathrm{WV}} T}$, is primarily temperature-dependent and independent of ambient pressure. Moreover, the adiabatic heating in the descending vortex pair does not depend on ambient pressure.

| Nr | $\hat{f}_{N,s}$ | $\tilde{f}_{N,s}$ | $z_{atm}$ | $\tilde{z}_{atm}$ / m | $z_{emit}$ / m | $\tilde{z}_{emit}$ / m |
|---|---|---|---|---|---|---|
| 1,2,3,4,5 | 0.06, 0.23, 0.6, 0.87, 0.97 | 0.05, 0.21, 0.59, 0.87, 0.98 | 164 | 163 | 249 | 250 |
| 6,7,8,9,10 | 0.0, 0.06, 0.22, 0.58, 0.86 | 0.0, 0.05, 0.22, 0.59, 0.87 | 85 | 87 | 249 | 250 |
| 11,12,13,14,15,16 | 0.05, 0.2, 0.57, 0.85, 0.96, 1.0 | 0.05, 0.19, 0.55, 0.86, 0.97, 1.0 | 164 | 163 | 546 | 541 |
| 17,18,19,20,21 | 0.08, 0.29, 0.68, 0.9, 0.98 | 0.08, 0.28, 0.68, 0.91, 0.99 | 85 | 87 | 546 | 541 |
| 22,23,24 | 0.06, 0.6, 0.97 | 0.05, 0.59, 0.98 | 164 | 163 | 249 | 250 |
| 25,26,27 | 0.06, 0.6, 0.97 | 0.05, 0.59, 0.98 | 164 | 163 | 249 | 250 |
| 28,29,30 | 0.0, 0.21, 0.86 | 0.0, 0.21, 0.87 | 85 | 87 | 249 | 250 |
| 31,32,33 | 0.0, 0.22, 0.86 | 0.0, 0.22, 0.87 | 85 | 87 | 249 | 250 |
| 34,35,36 | 0.2, 0.85, 1.0 | 0.19, 0.85, 1.0 | 164 | 163 | 546 | 541 |
| 37,38,39 | 0.2, 0.85, 1.0 | 0.19, 0.86, 1.0 | 164 | 163 | 546 | 541 |
| 40,41,42 | 0.08, 0.68, 0.98 | 0.08, 0.68, 0.99 | 85 | 87 | 546 | 541 |
| 43,44,45 | 0.08, 0.68, 0.98 | 0.08, 0.68, 0.99 | 85 | 87 | 546 | 541 |
| 46,47,48 | 0.0, 0.26, 0.94 | 0.0, 0.25, 0.95 | 164 | 163 | 249 | 250 |
| 49,50,51 | 0.0, 0.02, 0.73 | 0.0, 0.02, 0.74 | 85 | 87 | 249 | 250 |
| 52,53,54 | 0.02, 0.71, 1.0 | 0.01, 0.7, 1.0 | 164 | 163 | 546 | 541 |
| 55,56,57 | 0.0, 0.36, 0.96 | 0.0, 0.35, 0.97 | 85 | 87 | 546 | 541 |
| Simulations at higher ambient temperatures | | | | | | |
| 58,59,60,61,62 | 0.03, 0.14, 0.44, 0.79, 0.94 | 0.02, 0.13, 0.43, 0.8, 0.95 | 177 | 176 | 110 | 112 |
| 63,64,65,66,67 | 0.0, 0.01, 0.09, 0.3, 0.69 | 0.0, 0.0, 0.09, 0.31, 0.7 | 92 | 95 | 110 | 112 |
| 68,69,70,71,72 | 0.08, 0.28, 0.67, 0.9, 0.98 | 0.07, 0.27, 0.67, 0.9, 0.99 | 177 | 176 | 262 | 263 |
| 73,74,75,76,77 | 0.0, 0.07, 0.27, 0.65, 0.89 | 0.0, 0.07, 0.27, 0.66, 0.9 | 92 | 95 | 262 | 263 |
| 78,79,80,81,82 | 0.05, 0.21, 0.57, 0.86, 0.97 | 0.05, 0.2, 0.56, 0.86, 0.97 | 186 | 185 | 163 | 163 |
| 83,84,85,86,87 | 0.0, 0.03, 0.15, 0.46, 0.81 | 0.0, 0.03, 0.15, 0.47, 0.82 | 97 | 99 | 163 | 163 |
| 88,89,90,91,92 | 0.05, 0.19, 0.53, 0.84, 0.96 | 0.04, 0.17, 0.52, 0.84, 0.96 | 191 | 191 | 123 | 121 |
| 93,94,95,96,97 | 0.0, 0.02, 0.12, 0.38, 0.76 | 0.0, 0.02, 0.12, 0.39, 0.77 | 99 | 102 | 123 | 121 |
| 98,99,100 | 0.04, 0.53, 0.96 | 0.04, 0.52, 0.96 | 191 | 191 | 123 | 121 |
| 101,102,103 | 0.05, 0.53, 0.96 | 0.04, 0.52, 0.97 | 191 | 191 | 123 | 121 |
| 104,105,106 | 0.0, 0.12, 0.76 | 0.0, 0.12, 0.77 | 99 | 102 | 123 | 121 |
| 107,108,109 | 0.0, 0.12, 0.76 | 0.0, 0.12, 0.77 | 99 | 102 | 123 | 121 |
| 110,111,112,113,114 | 0.04, 0.18, 0.52, 0.83, 0.96 | 0.04, 0.16, 0.5, 0.83, 0.96 | 195 | 194 | 102 | 99 |
| 115,116,117,118,119 | 0.0, 0.01, 0.1, 0.34, 0.73 | 0.0, 0.01, 0.1, 0.35, 0.74 | 101 | 104 | 102 | 99 |
| Simulations with A320/B737-like aircraft | | | | | | |
| 120 | 0.72 | 0.72 | 164 | 163 | 176 | 183 |
| 121,122,123 | 0.13, 0.64, 0.96 | 0.12, 0.64, 0.97 | 177 | 176 | 76 | 79 |
| 124,125,126 | 0.03, 0.21, 0.78 | 0.03, 0.21, 0.8 | 92 | 95 | 76 | 79 |
| 127,128,129 | 0.2, 0.76, 0.99 | 0.19, 0.76, 0.99 | 177 | 176 | 185 | 186 |
| 130,131,132 | 0.07, 0.39, 0.9 | 0.06, 0.4, 0.91 | 92 | 95 | 185 | 186 |
| 133,134,135 | 0.17, 0.72, 0.98 | 0.16, 0.72, 0.98 | 186 | 185 | 114 | 113 |
| 136,137,138 | 0.04, 0.29, 0.85 | 0.04, 0.29, 0.86 | 186 | 185 | 114 | 113 |
| 139,140,141 | 0.16, 0.71, 0.97 | 0.15, 0.7, 0.98 | 191 | 191 | 85 | 83 |
| 142,143,144 | 0.04, 0.26, 0.82 | 0.03, 0.26, 0.83 | 99 | 102 | 85 | 83 |
| 145,146,147 | 0.16, 0.7, 0.97 | 0.15, 0.69, 0.98 | 195 | 194 | 70 | 67 |
| 148,149,150 | 0.03, 0.24, 0.81 | 0.03, 0.24, 0.82 | 101 | 104 | 70 | 67 |

**Table S1.** List of parametrized survival fractions derived with length scales that are computed via the numerical (Eqs. (6) and (7)) or the analytical method (Eqs. (A2) and (A3)), denoted with a tilde, and the corresponding length scales. Rows with three, five, or six simulations correspond to sets where the $N_0$-scaling factors 100, 1, 0.01; 100, 10, 1, 0.1, 0.01; or 1000, 100, 10, 1, 0.1, 0.01 are applied, respectively.